# Cosmc controls B cell homing

Junwei Zeng[1], Mahmoud Eljalby [2], Rajindra P. Aryal [1], Sylvain Lehoux[1], Kathrin Stavenhagen[1], Matthew R. Kudelka[1,3], Yingchun Wang[3], Jianmei Wang[3], Tongzhong Ju[3,5], Ulrich H. von Andrian [2,4] & Richard D. Cummings [1✉]

The molecular mechanisms regulating lymphocyte homing into lymph nodes are only partly understood. Here, we report that B cell-specific deletion of the X-linked gene, *Cosmc*, and the consequent decrease of protein O-glycosylation, induces developmental blocks of mouse B cells. After transfer into wild-type recipient, *Cosmc*-null B cells fail to home to lymph nodes as well as non-lymphoid organs. Enzymatic desialylation of wild-type B cells blocks their migration into lymph nodes, indicating a requirement of sialylated O-glycans for proper trafficking. Mechanistically, *Cosmc*-deficient B cells have normal rolling and firm arrest on high endothelium venules (HEV), thereby attributing their inefficient trafficking to alterations in the subsequent transendothelial migration step. Finally, *Cosmc*-null B cells have defective chemokine signaling responses. Our results thus demonstrate that *Cosmc* and its effects on O-glycosylation are important for controlling B cell homing.

[1] Department of Surgery, Beth Israel Deaconess Medical Center, Harvard Medical School, Boston, MA, USA. [2] Department of Microbiology & Immunobiology, Harvard Medical School, Boston, MA, USA. [3] Department of Biochemistry, Emory University, Atlanta, GA, USA. [4] The Ragon Institute of MGH, MIT & Harvard, Cambridge, MA, USA. [5]Present address: Office of Biotechnology Products, Center for Drug Evaluation and Research, U. S. Food and Drug Administration, Silver Spring, MD 20993, USA. ✉email: rcummin1@bidmc.harvard.edu

Naïve lymphocytes continuously patrol the body in search of cognate antigens and readily mount immune responses. Thus, constitutive circulation of lymphocytes between blood and lymphoid systems is essential for immune surveillance. Naïve lymphocytes enter lymph nodes through a complex and partly understood process that begins with a series of molecular interactions requiring glycans on endothelial cells of lymph nodes that are recognized by L-selectin on lymphocytes[1–3]. This recognition by L-selectin of peripheral node addressins (PNAds) on high endothelial venules (HEV) mediates initial tethering and rolling of lymphocytes[2,4–6]. Subsequent chemokine signaling through G protein-coupled receptors, e.g. CCR7 and its ligands CCL19 and CCL21, activate lymphocyte integrins, leading to firm arrest, and finally the diapedesis of the adherent cells into the lymph node[7]. It is not clear, however, whether the glycans on the lymphocytes themselves are important in homing.

The potential importance of lymphocyte glycans in homing was suggested by the seminal studies of Gesner and Ginsberg[8,9]. Using a radioactivity-based assay, their findings suggested that glycosidase treatment of intact lymphocytes decreased their homing to lymph nodes in recipient rats. The nature and functions of the lymphocyte glycoconjugates implicated in this process remain unknown.

Many studies have demonstrated that mucin-type O-glycans, characterized by extended modifications of the core 1 O-glycan structure Galβ1-3GalNAc1-α-Ser/Thr, which are expressed on granulocytes and activated T cells, are important in many aspects of leukocyte trafficking in inflammation, through interactions with selectins[10]. The extension of O-glycans in normal cells occurs by addition of other sugars, including N-acetylglucosamine, fucose and sialic acid[11]. There is scant information, however, about the roles of lymphocyte glycans in homing, but some studies suggest the possibility that O-glycosylation might be important[12–14].

The presence of galactose residues in core 1 O-glycans requires expression of a single enzyme T-synthase, encoded by *T-synthase (C1GalT1)*, which in the Golgi apparatus converts Tn antigen (CD175) GalNAcα1-Ser/Thr to the ubiquitous core-1 O-glycan[15,16]. The formation of active T-synthase requires a dedicated molecular chaperone termed Cosmc in the endoplasmic reticulum[17]. *Cosmc (C1GalT1C1)* is encoded on the X chromosome (human Xq24, mouse Xc3) and systemic deletion of either *T-synthase* or *Cosmc* leads to an embryonic lethality[18,19]. Conditional deletion of *Cosmc* in hematopoietic and endothelial cells results in severe pathology that leads to embryonic death; interestingly, surviving mice suffer macrothrombocytopenia and perinatal hemorrhage and die within a few months[20].

With the long-term goal of understanding the role of O-glycans on B cell biology, here we generate and characterize the murine B cell-specific *Cosmc* KO mice, which have specifically blocked extension of O-GalNAc-type O-glycans on glycoproteins of B cells. Our subsequent analyses demonstrate a critical role of *Cosmc* and extended O-glycans in B cell development and homing.

## Results

**Reduced B cells in B cell-specific *Cosmc*-KO lymph nodes**. To directly address the potential roles of O-glycans in B cell homing, we engineered a targeted deletion of *Cosmc* in B cells by crossing the *LoxP*-flanked *Cosmc* mice with *Mb1*-Cre mice to generate B cell-specific *Cosmc*-knockout (BC-*Cosmc*KO) mice[21] and further confirmed *Cosmc* deletion in B220+ B cells (Supplementary Fig. 1A, B). Additionally, we analyzed surface expression of the Tn antigen (CD175), an abnormal glycan structure that can arise from dysfunctional *Cosmc*, on splenic B cells in BC-*Cosmc*KO mice. We found that >96% of splenic B cells express Tn antigen, demonstrating efficient *Cosmc* knockout (Supplementary Fig. 1C). The BC-*Cosmc*KO mice appeared outwardly normal and healthy at the time of experimentation. However, the BC-*Cosmc*KO mice exhibited splenomegaly and the wet spleen mass was ~60% greater than that of WT (Fig. 1a).

Cellularity analysis by flow cytometry indicated reduced B cell numbers by 40% from the BC-*Cosmc*KO mice in the spleen and bone marrow (BM) (Fig. 1b, c), compared to WT. By contrast, we observed a remarkable decrease of B cells in peripheral blood lymphocytes (PBL) and peripheral lymph nodes (PLNs) (Fig. 1d), indicating profound migration defects in *Cosmc*-deficient B cells. To further examine whether the B cell reduction in PLNs of BC-*Cosmc*KO mice is extended to gut associated lymphoid tissues (GALTs), we analyzed the mesenteric lymph nodes (MLNs) and Peyer's patches (PPs). Similar to PLNs, we observed a marked reduction of B cells in the MLNs and PPs of BC-*Cosmc*KO mice (Fig. 1e). We also examined T lymphocyte subsets in peripheral lymphoid organs in BC-*Cosmc*KO mice. Compared to littermate wild-type control, both CD4+ and CD8+ T cells were increased in spleen, which likely contribute to the splenic enlargement (Supplementary Fig. 1D). Additionally, we observed pronounced reduction of B cells in the lung and liver of BC-*Cosmc*KO mice (Fig. 1f). We analyzed the spleen from both BC-*Cosmc*KO and WT mice using immunofluorescence staining and observed normal B cell follicles (Fig. 1g). Consistent with our flow cytometry data, we observed significant reduction of CD19-immunostained B cells in the peripheral and mesenteric lymph nodes of BC-*Cosmc*KO mice (Fig. 1h–i). We observed rudimentary PPs in the small intestine of the BC-*Cosmc*KO mice as compared to WT (Fig. 1j) which is consistent with previous studies suggesting a role for B cells in PPs organogenesis[22,23].

**Altered B cell development in BC-*Cosmc*KO mice**. To investigate the role of *Cosmc* in B cell development, we analyzed the B cell subsets from the BM and the spleen of both wild-type and BC-*Cosmc*KO mice using flow cytometry (Fig. 2). Notably, we found that the major defects were observed in B cell progenitors after the pro-B cell stage (Hardy fraction B), where *Mb1-Cre* becomes active, in bone marrow of the BC-*Cosmc*KO mice (Fig. 2a, b). We observed >90% reduction of both small pre-B cell (Hardy fraction D) and recirculating mature B cell (Hardy fraction F), and also on the other side we observed a 3.5-fold increase of immature B cells (Hardy fraction E) in the BC-*Cosmc*KO mice. In the spleen, both the IgM+IgD+ and IgM+IgD− B cells were reduced (Fig. 2c). The marginal zone (MZ) B cells were increased >twofold, with a concomitant reduction of follicular (FO) B cells in the BC-*Cosmc*KO mice (Fig. 2d). Of note, we observed enhanced expression of CD21 on splenic B cells, as well as IgM and IgD on B cells, complicating the demarcation of the B cell subsets using these markers (Fig. 2a–d). The *Cosmc*-deficient mature B cells were substantially reduced in the bone marrow, compared to those in the spleen, indicating a possibly impaired recirculation of mature B cells back to the bone marrow. Collectively, these data demonstrate that *Cosmc* mutation in B cells alters their development in both BM and spleen.

BC-*Cosmc*KO mice exhibited reduced B cell numbers in bone marrow and peripheral lymphoid tissues. Paradoxically, BC-*Cosmc*KO mice exhibited increased levels of total IgM in serum, which may reflect the increased numbers of MZ B cells (Fig. 2e). Meanwhile, BC-*Cosmc*KO mice demonstrated reduced serum IgA level, an immunoglobulin isotype that is primarily produced by intestinal B cells. BC-*Cosmc*KO mice had scarce B cells in Peyer's patches, which likely contributes to the decreased amount of serum IgA. In addition, dynamic changes in IgG isotypes level

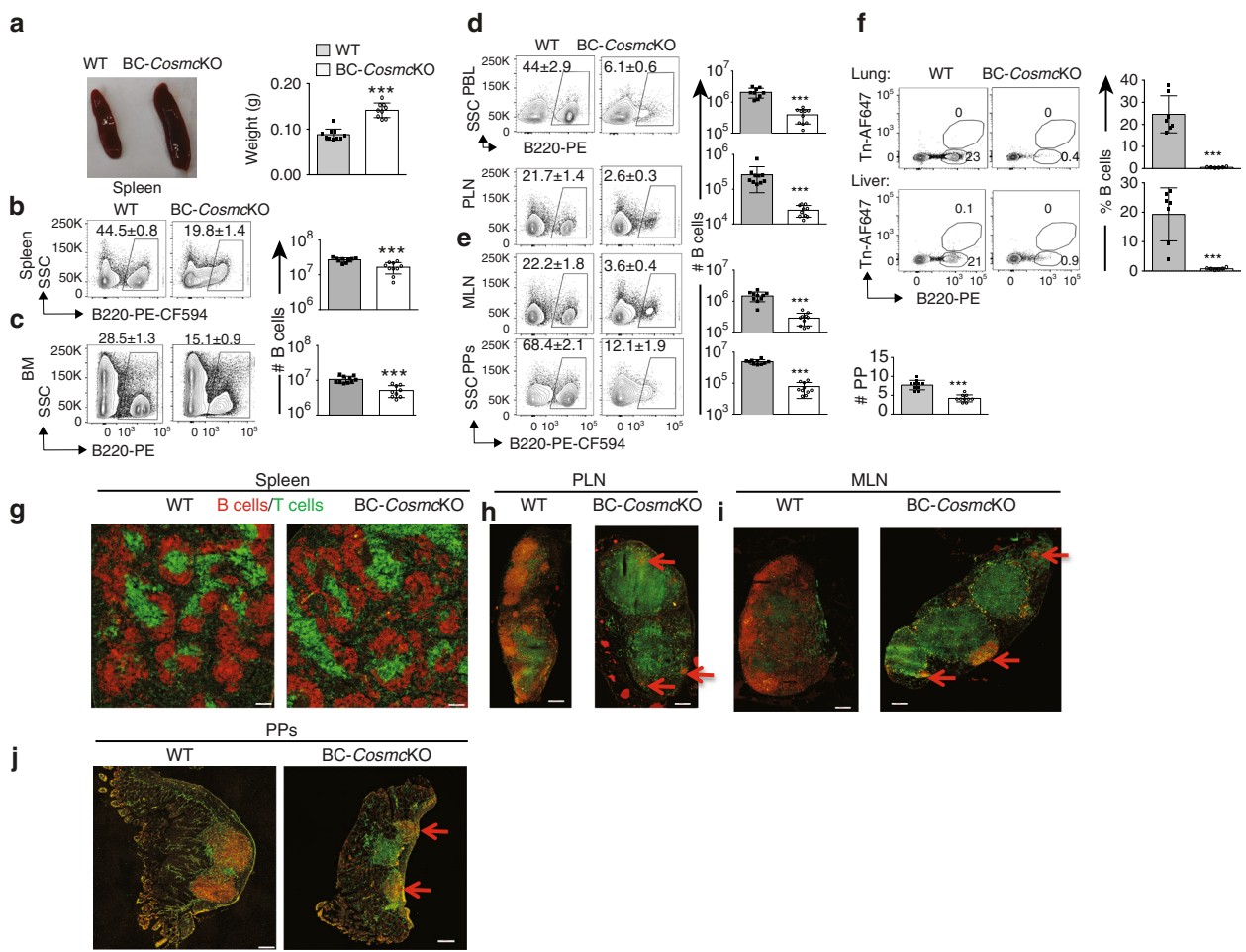

**Fig. 1 Substantial reduction of B cells number in BC-*Cosmc*KO mice lymphoid organs.** Both WT littermate control and BC-*Cosmc*KO mice at 8 weeks old were used in experiments. Each symbol (black square and open circle for WT and BC-*Cosmc*KO, respectively) represents an individual mouse. **a** representative photograph of spleens of WT littermate control (n = 10) and BC-*Cosmc*KO (n = 9) mice. Statistical analysis of spleen weight is shown in bar graph and p value < 0.0001. **b–f** Frequencies and numbers of B220[+] B cells were determined in indicated tissues by flow cytometry (n = 10 for WT, n = 10 for BC-*Cosmc*KO for all tissues, except for n = 9 for WT in PBL, and n = 9 for BC-*Cosmc*KO in BM): (**b**) spleen, p value < 0.0001, (**c**) bone marrow (BM), from two femurs, p value < 0.0001, (**d**) PBL per ml, and PLNs, both p values < 0.0001. **e** Mesenteric lymph node (MLN) and Peyer's Patches (PPs), the numbers of PPs, and all of p values < 0.0001, and (**f**) Co-stained with antibody against abnormal O-glycan structure (Tn) in lung, p value < 0.0001 and liver, p value = 0.0004. Data are presented as average ±SD of each genotype. **g–j** Representative immunofluorescence staining of the cryostatic sections (n = 5) of spleen (**g**), PLN (**h**), MLN (**i**), and PPs (**j**), stained with anti-CD19-PE for B cells and anti-Thy1.2-FITC for T cells, and acquired at ×10 magnification. Scale bar represents 200 μm. Red arrow points to the location of the B cells. Unpaired two-tailed Student's t tests were performed to determine statistical significance with *** denoting p < 0.001. Source data are provided as a Source Data file.

were observed, with increased levels of IgG2b, IgG2c, and a surprisingly marked reduction of IgG3 (Fig. 2e).

**Cosmc controls B cell homing to LNs and non-lymphoid organs.** We were intrigued by the disproportionate reduction of resident B cells number in the spleen, PLNs, and PPs of the BC-*Cosmc*KO mice as compared to WT. To determine whether homing of B cells into those tissues is affected, we conducted adoptive transfer experiments. We isolated splenic cells from either BC-*Cosmc*KO mice or littermate controls, labeled them with CellTrace violet, and co-injected into WT recipient mice with CFSE-labeled WT splenocytes as an internal control. Mice were sacrificed at 2 or 20 h post-injection, and flow cytometry was used to determine the frequencies of their appearance in PBL, spleen, MLNs, PPs, and PLNs. Remarkably, at 2 h, there were few *Cosmc*-deficient B cells in LNs and PPs (Fig. 3a), as we observed less than 2% of *Cosmc*-deficient B cells migrated to PLNs, MLNs, and PPs, as compared to control B cells. After

20 h, the accumulation of *Cosmc*-deficient B cells remained significantly reduced, with only 3% in PLNs, and less than 1% in both MLNs, and PPs (Fig. 3b), compared to control B cells. We also observed that a negligible number of the *Cosmc*-deficient B cells had migrated into the lung and liver (Fig. 3c). We obtained similar results when we used BC-*Cosmc*KO mice as recipients (Supplementary Fig. 2A, B). Collectively, these data demonstrate that *Cosmc* is essential for normal B cell migration to both lymphoid and non-lymphoid organs, in a cell-intrinsic manner.

**Glycan profiling of B cells.** To gain insights into the nature of glycans on *Cosmc*-deficient B cells, we first analyzed the released Asn-linked oligosaccharides (N-glycans) from purified WT and *Cosmc*-deficient B cells using mass spectrometry. Our analysis of the N-glycan profile showed no significant differences compared to WT (Supplementary Fig. 3A, B), indicating that the deletion of *Cosmc* in B cells does not affect N-glycosylation pathways. In

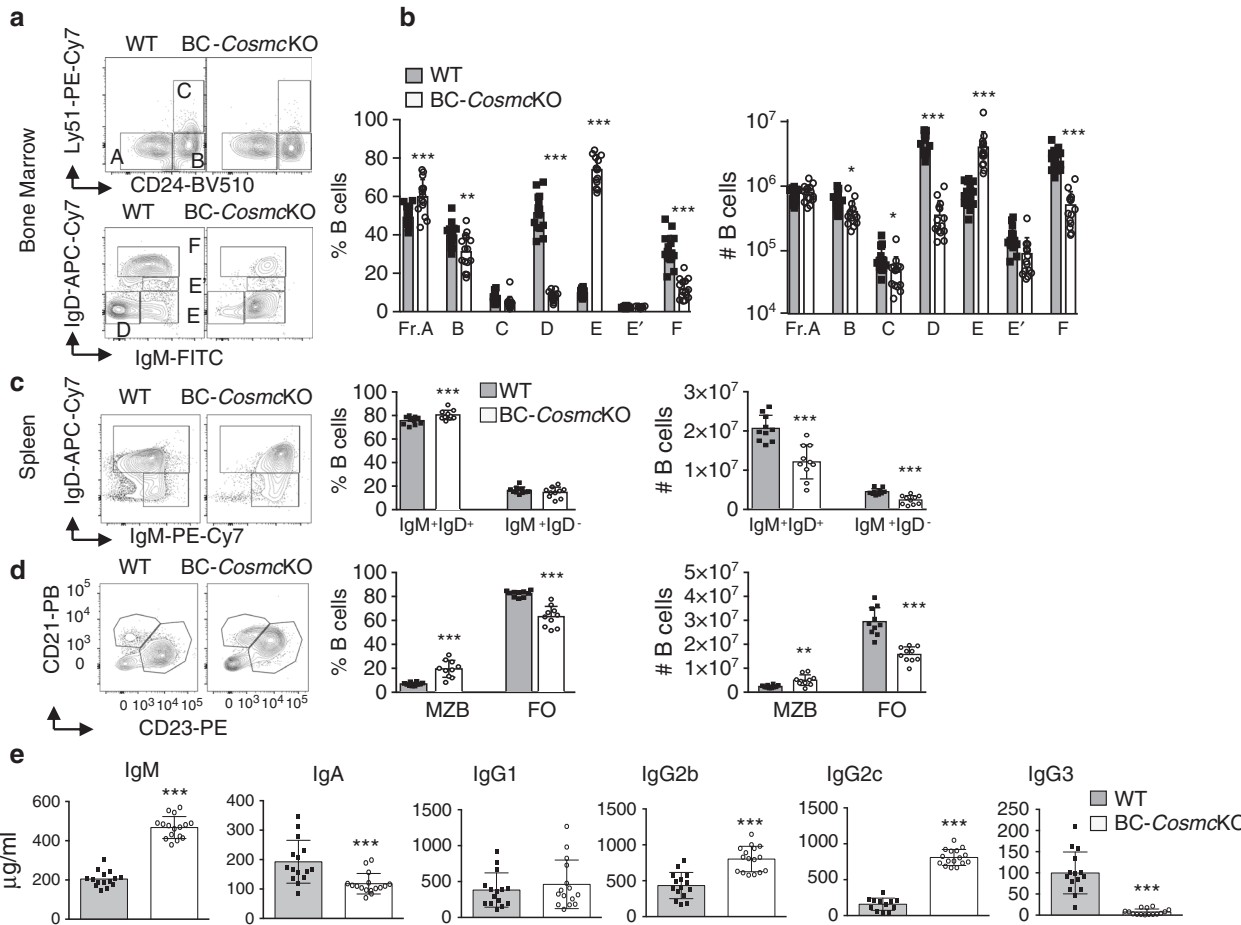

**Fig. 2 _Cosmc_ is required for B cell development.** Single cell suspensions were prepared from both bone marrow and spleen of WT and BC-_Cosmc_KO mice and stained with indicated antibodies. Each symbol (black square and open circle for WT and BC-_Cosmc_KO, respectively) represents an individual mouse. Representative flow cytometric contour plots and numbers of B cell subsets were shown in (**a**, **b**) bone marrow (n = 16 for WT and n = 15 for BC-_Cosmc_KO mice), in %B cells bar graphs: p values of fraction (**a**) 0.0003, (**b**) 0.0032, (**c**) 0.0717, (**d**) <0.0001, (**e**) <0.0001, (**e'**): 0.7302, (**f**) <0.0001, in #B cells bar graphs: p values of fraction (**a**) 0.2217, (**b**) 0.0167, (**c**) 0.0148, (**d**) <0.0001, (**e**) <0.0001, (**e'**): 0.0093, (**f**) <0.0001, and (**c**, **d**) spleen (n = 13 for WT and n = 14 for BC-_Cosmc_KO mice for **c**, and n = 11 for WT and n = 10 BC-_Cosmc_KO mice for **d**. In %B cells bar graphs of **c**: p values of IgM⁺IgD⁺ = 0.0003, of IgM⁺IgD⁻ = 0.5633. In #B cells bar graphs of **c** p values of IgM⁺IgD⁺ < 0.0001, of IgM⁺IgD⁻< 0.0001. In %B cells bar graphs of **d**: p values of MZB < 0.0001, of FO < 0.0001. In #B cells bar graphs of **d**: p values of MZB = 0.0013, of FO < 0.0001. Hardy's gating schemes were used to measure B cells at different developmental stage (**a**), with top row gated on B220⁺CD43⁺ cells, and bottom row gated on B220⁺CD43⁻ cells. **e** Serum from naïve BC-_Cosmc_KO mice and WT littermate control (for IgM, IgA, and IgG2b, n = 15 for WT and n = 16 for BC-_Cosmc_KO; for IgG1 and IgG3, n = 15 for both groups; for IgG2c, n = 12 for WT and n = 16 for BC-_Cosmc_KO) were assessed for indicated immunoglobulin isotypes levels by sandwich ELISA with appropriate immunoglobulin standards. Each symbol represents the datum from an individual mouse. For IgM, p value < 0.0001, for IgA, p value = 0.0003, for IgG1, p value = 0.4629, for IgG2b, p value < 0.0001, for IgG2c, p value < 0.0001, for IgG3, p value < 0.0001. Data are presented as average ±SD of each genotype. Unpaired two-tailed Student's t tests were performed to determine statistical significance with *** denoting p < 0.001, **p < 0.01. Source data are provided as a Source Data file.

parallel studies, we also analyzed glycosylation of mouse IgG. IgG N-glycopeptide analysis revealed very similar glycan profiles among all IgG subtypes with minor differences in IgG sialylation (Supplementary Fig. 4A–D). Importantly, we observed that B cells derived from the BC-_Cosmc_KO mice lacked extended O-glycans in their glycoproteins as compared to WT (Fig. 4a, b). This was confirmed on a protein-specific level in our analysis of the hinge-region of IgG2b. Thr104 (UniProt annotation) was identified to be partially O-glycosylated (Supplementary Fig. 5A, B), which is consistent with a previous report[24]. While IgG2b from WT mice sera expresses mono- and disialylated core 1 O-glycans, the IgG2b from BC-_Cosmc_KO mice shows exclusively the Tn antigen (Fig. 4c). These results demonstrate that _Cosmc_ deletion does not affect N-glycan structures, but causes the loss of extended O-glycans, resulting in the expression of the Tn antigen on B cells. Also consistent with a previous study[25], N-glycans from B cells

include biantennary complex-type N-glycans capped with the sialic acid Neu5Gc, as well as Neu5Ac (Supplementary Fig. 3A). Moreover, we identified abundant high-mannose-type N-glycans, as well as poly-N-acetyllactosamine-containing glycans (–3Galβ1-4GlcNAcβ1–)ₙ (Supplementary Fig. 3A). Notably, after neur-aminidase (sialidase) treatment, the binding of PNA, which binds to the core 1 disaccharide Galβ1-3GalNAcα1-Ser/Thr, was enhanced on both WT B and T cells, as expected (Supplementary Fig. 6A, B). By contrast, the binding of _Maackia amurensis_ lectin-II (MAL-II), which is specific for α2-3-linked sialic acid on the core 1 disaccharide, as well as the binding of _Sambucus nigra_ agglutinin (SNA), specific for α2-6-linked sialic acids, were decreased in both WT B and T cells (Supplementary Fig. 6A, B). Together, these results demonstrate that glycoproteins of WT murine B cells express extended and sialylated O-glycans, which are lacking on BC-_Cosmc_-deficient B cells.

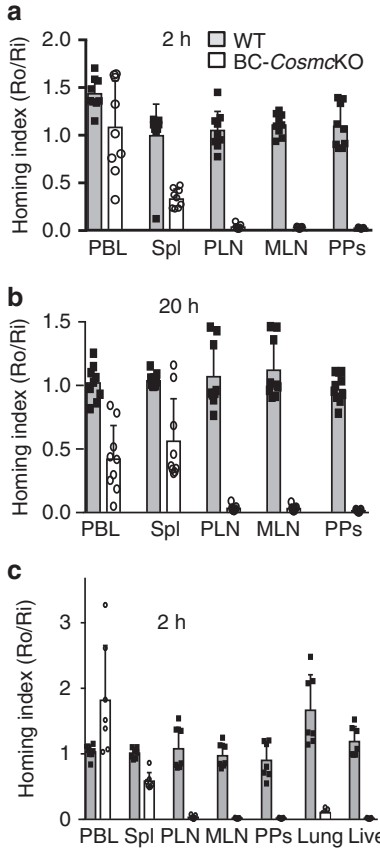

**Fig. 3 *Cosmc* deficiency in B cells blocks B cell homing.** Splenic cells from WT and BC-*Cosmc*KO mice were labeled with CellTrace Violet then transferred to WT mice separately ($n = 9$), or mixed and co-transferred to WT ($n = 7$) recipient mice with CFSE-labeled WT splenocytes as an internal control. Each symbol (black square and open circle for WT and BC-*Cosmc*KO, respectively) represents an individual mouse. Homed donor Tn⁻ or Tn⁺ B220⁺ (or CD19⁺) B cells were harvested at 2 (**a**, **c**) or 20 h (**b**) and analyzed by flow cytometry. The homing index was calculated as the [percentage of dye⁺ Tn⁻ or Tn⁺ B cells] $_{tissue}$/[percentage of internal control dye⁺ B cells] $_{tissue}$ ratio to the input ratio. Data are presented as average ± SD of each genotype with each individual value plotted. Source data are provided as a Source Data file.

**Removing sialic acid on lymphocytes blocks migration to LNs.** The above results demonstrate that extended O-glycans are required for proper B cell homing to lymph nodes. As O-glycans are frequently capped with sialic acid, we treated lymphocytes with neuraminidase in order to identify whether sialic acids on lymphocytes play a role their homing to lymph nodes (Fig. 4a, c). This treatment, however, is not specific for O-glycans, as it releases sialic acids from both O- and N-glycans, and there is no available neuraminidase that specifically can desialylate O-glycans (Supplementary Fig. 6A, B). We adoptively transferred a preparation of neuraminidase-treated and dye-labeled bulk WT splenocytes into WT recipient mice and examined their distribution within diverse organs using flow cytometry. We observed that in WT recipients, PBS-treated WT cells were distributed as expected into lymph nodes and other organs. However, neuraminidase-treated WT splenocytes accumulated in the liver; strikingly, the majority of these cells were Thy1.2⁺ T cells, which were 5-fold increased, compared to untreated T cells (Fig. 4d, Supplementary Fig. 6C). Remarkably, neither neuraminidase-treated B cells nor control B cells were targeted to the liver, as the neuraminidase-treated B cells showed

substantially impaired migration to the liver, compared to control B cells (Fig. 4d). Both neuraminidase-treated T and B cells showed significantly reduced accumulation in PLNs, PPs, and lung (Fig. 4e, f, Supplementary Fig. 6D). These results demonstrate that surface sialic acid on lymphocytes is crucial for T and B cell maintenance in the periphery. More importantly, based on this result and those above with BC-*Cosmc*KO mice, we conclude that sialylated O-glycans are required for B cell homing.

**Intravital microscopy of *Cosmc*-deficient B cell homing.** To directly assess whether other aspects of B cell homing, such as cell rolling or adhesion to endothelium, which largely is known to depend on L-selectin, might be defective in the *Cosmc*-deficient B cells, we performed intravital microscopy (IVM) in inguinal lymph nodes. For the five discrete venular orders examined, the rolling and sticking fraction of B cells in HEVs was not affected in BC-*Cosmc*KO mice (Fig. 5a, b, representative videos are shown in Supplementary Movies 1 and 2). Interestingly, when we measured the rolling velocity ($V_{roll}$), which reflects cell movement while adhesively contacting the HEV surface, we found that the median $V_{roll}$ of *Cosmc*-deficient B cells was increased to 122.4 μm/s in order III HEVs, 68% higher than it was in WT control (Fig. 5c). $V_{roll}$ has been reported to be sensitive to L-selectin expression level, with higher expression of CD62L resulting in lowered $V_{roll}$[12]. These data suggest that *Cosmc* contributes to some extent to the rolling interaction after tethering, but these relatively modest effects are unlikely to contribute to the major defects in homing of *Cosmc*-deficient B cells.

Based on the above results that *Cosmc*-deficient B cells lack major defects in rolling and arrest, we considered the possibility that the cells may be defective in terminal steps of transmigration and diapedesis. Lymphocytes enter the lymph nodes through HEVs, which involves a cascade of molecular events between lymphocytes and HEVs[1–3,7]. To dissect the molecular mechanisms of the defective migration of *Cosmc*-deficient B cells, we examined specific factors that might be affected in the cells. Using flow cytometry, we analyzed the expression levels of several known migration-related molecules, such as L-selectin (CD62L)[26,27] and integrin α4β7[28,29], which participate in mediating lymphocyte homing. Interestingly, we observed somewhat higher expression levels of L-selectin, β7 and α4β7 on *Cosmc*-deficient B cells as compared to WT littermate control (Fig. 5d). We also cytometrically examined the surface expression of several chemokine receptors that are known to contribute to lymphocyte migration[30–32]. Notably, the exact chemokine receptor(s) required for B cell homing to lymph nodes at the genetic level are not well understood, and there may be functional redundancy. We observed higher expression levels of CXCR4, CXCR5, CCR7 on *Cosmc*-deficient B cells as compared to WT (Fig. 5e). It is known that CCR7, a receptor for both CCL19 and CCL21, has several sites of O-glycosylation in its N-terminal domain[33,34]. Additionally, a previous study demonstrated that sialylation and O-glycosylation of CCR5 in its extracellular domain is important for normal chemokine binding[35]. To explore this further, we immunoprecipitated mouse CCR7 and then blotted with PNA lectin, which binds to core 1 O-glycans. The binding of PNA is consistent with the presence of O-glycans on murine CCR7 (Supplementary Fig. 7).

Thus, we reasoned that chemokine receptors in *Cosmc*-deficient B cells, lacking sialylated extended O-glycans, might have altered chemotactic responsiveness. To test this possibility, we explored whether *Cosmc*-deficient B cells exhibited defective chemotaxis in a transwell migration assay. Compared to their WT counterpart, the response of *Cosmc*-deficient B cells to CXCL13 was normal, whereas it was significantly reduced for CCL21 and

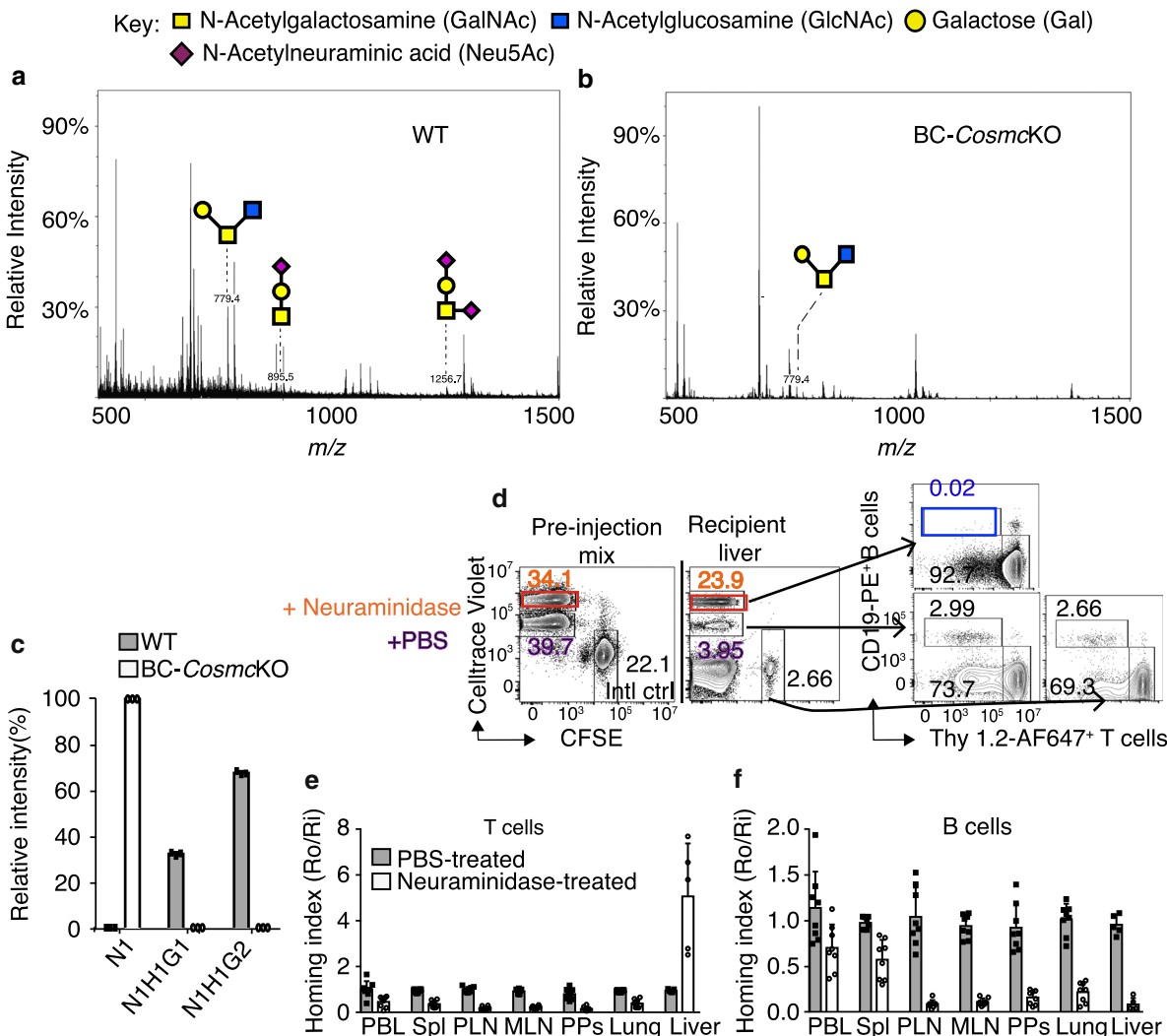

**Fig. 4 Sialylation differentially regulates T and B cell trafficking.** O-Glycans (**a**, **b**) were extracted from splenic B cells that were purified from WT (**a**) and BC-*Cosmc*KO (**b**) mice. The released glycans were subjected to mass spectrometric analysis. The peaks with annotated glycan structures are listed. Other peaks are background noise, not matching to any glycan mass. **c** IgG2b hinge-region O-glycosylation characterization in WT and BC-*Cosmc*KO serum. The O-glycan distribution of the IgG2b hinge-region tryptic peptide (K)LEPSGPISTINPCPPCK. Thr104 (UniProt annotation) was identified to be O-glycosylated. Glycan compositions are indicated with H—hexose; N—N-acetylhexosamine; G— N-glycolylneuraminic acid. **d–f** Single cell suspension of splenocytes from WT mice were labeled with CellTrace Violet or CFSE and treated with neuraminidase. The preparation of the splenocytes was co-injected into recipient mice (*n* = 8, except for *n* = 5 for liver). Lymphoid tissues were harvested at 1 h after transfer. **d** Representative flow cytometric dot plots show input cells ratio before injection and after transfer. Neuraminidase in orange rectangle, and numbers in orange (percentage of total input) indicated donor cells treated with Neuraminidase. PBS in purple numbers (percentage of total input) indicated donor cells treated with PBS. Blue rectangle and numbers indicated neuraminidase-treated and transferred B cell population that recovered from the indicated tissue of recipients. **e**, **f** Homing indices were calculated based on the percentage of homed T and B cells. The homing index was calculated as the [percentage of dye+CD19+ B cells or Thy1.2+ T cells] $_{tissue}$/ [percentage of internal control CFSE+B cells or T cells] $_{tissue}$ ratio to the input ratio. Each symbol (black square and open circle for WT and BC-*Cosmc*KO, respectively) represents an individual mouse. Data are presented as average ±SD of each genotype with each individual value plotted. Source data are provided as a Source Data file.

CXCL12 (Fig. 5f). These results demonstrate that *Cosmc*-deficient B cells have impaired chemokine responsiveness.

## Discussion

Multiple factors have been reported in mediating lymphocyte homing to lymphoid organs[1–3], yet little is known about the role of O-glycans in B cell homing. Here we present our discovery that the deletion of *Cosmc* in murine B cells causes a loss of core-1 O-glycans, associated with a marked reduction of B cells in PLNs, PPs, and non-lymphoid organs. We found that the loss of *Cosmc* in B cells disturbs B cell development, and ablates their homing to PLNs, GALTs, and non-lymphoid organs. Our results

demonstrate that the rolling and firm attachment of *Cosmc*-deficient B cells in the blood venules were comparable to those of WT B cells, suggesting that extended O-glycans on B cells are not required for those steps. However, *Cosmc*-deficient B cells exhibit defective responses to chemokines in vitro. Together, these data demonstrate that the transmigration of B cells into lymph nodes requires a functional *Cosmc* and extended O-glycans. The evidence that the trafficking pattern of *Cosmc*-deficient B cells phenocopied that of desialylated B cells supports a role for sialylated O-glycans on B cells in normal B cell homing.

We also demonstrated that sialidase treatment of lymphocytes impairs their migration to lymph nodes and PPs, and causes them

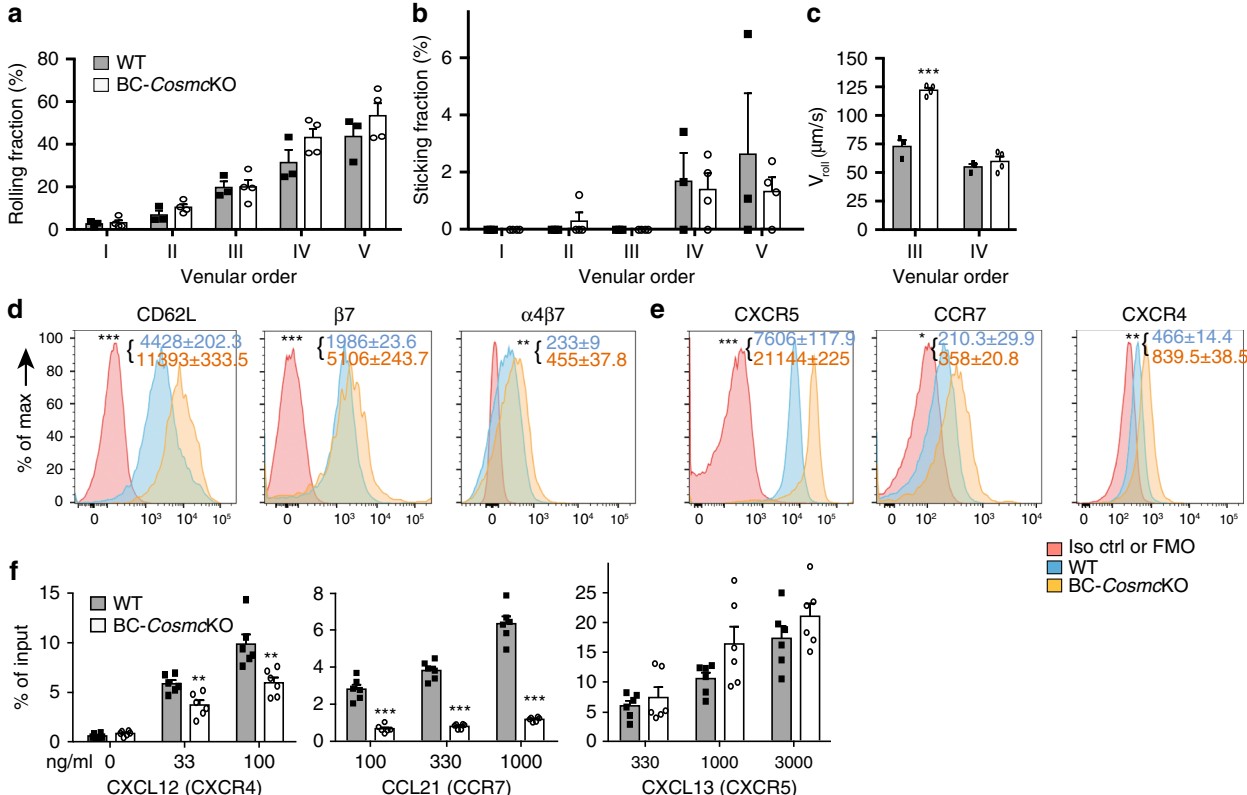

**Fig. 5 Intravital microscopy of *Cosmc*-deficient B cells. a–c** For IVM, purified B cells from WT and BC-*Cosmc*KO mice were labeled with calcein and injected into the left femoral artery catheter of the inguinal lymph node. Data from three independent experiments (*n* = 3) were analyzed using two-way analysis of variance (ANOVA) and presented as mean ± SEM, *** denoting *p* < 0.001. **a** Rolling fraction was measured as the percentage of rolling cells in the total flux of cells in each HEV. *p* values for venular order I, II, III > 0.9999, *p* value for venular order IV = 0.2154, *p* value for venular order IV = 0.4488. **b** Sticking fraction was determined as the percentage of adherent cells in the total lymphocyte flux through each HEV. p values for venular order I, II, III, IV > 0.9999, *p* value for venular order *V* = 0.9682. **c** Rolling velocity of B cells that rolled in order III and IV was calculated and quantified. *p* values for venular order III < 0.0001, *p* values for venular order IV = 0.7957. **d, e** Surface expression levels of (**d**) lymphocyte homing-related molecules, and (**e**) chemokine receptors on splenic B cells of WT control and BC-*Cosmc*KO mice. Splenic B cells were stained with indicated antibodies or isotype matched antibodies. Representative histogram plots comparing molecule expressions by mean fluorescence intensity (MFI) on B cells from both groups of mice. Histogram in pink: antibody isotype control or FMO, fluorescence minus one control. Histogram in blue: WT and Histogram in yellow: BC-*Cosmc*KO. *p* values for CD62L, β7, and α4β7 in **d** are < 0.0001, 0.0002, and 0.0046. *p* values for CXCR5, CCR7, and CXCR4 in **e** are < 0.0001, 0.0154, and 0.0016. Data are presented as average ±SEM of each genotype and represent a representative experiment from three independent experiments (*n* = 2 or 3 for WT and *n* = 3 for BC-*Cosmc*KO mice in each experiment) (**f**) Chemotaxis of *Cosmc*-deficient B220+ B cells compared with cells from littermate controls. Percentages of input cells that were found in lower well are shown. In the bar graphs on left side, *p* values for media only (0) is 0.0911, for 33 ng/ml of CXCL12 is 0.0048, and for 100 ng/ml of CXCL12 is 0.006. In the bar graphs in the middle, p values for all concentrations of CCL21 are < 0.0001. In the bar graphs on right side, *p* values for 330 ng/ml of CXCL13 is 0.4864, and for 1000 ng/ml of CXCL13 is 0.0898, *p* value for 3000 ng/ml of CXCL13 is 0.241. Data are presented as an average ±SD) of duplicate of each genotype and from three independent experiments (*n* = 3). Each symbol (black square and open circle for WT and BC-*Cosmc*KO, respectively) represents an individual mouse. For **d–f**, unpaired two-tailed student's *t* tests were performed to determine statistical significance with *** denoting *p* < 0.001, **p* < 0.01, **p* < 0.05. Source data are provided as a Source Data file.

to accumulate in the liver, consistent with earlier studies on the effects of glycosidase treatments of lymphocytes[8,9]. Strikingly, we found that most of the enriched desialylated lymphocytes in the liver are T cells, with only trace numbers of B cells (Fig. 4). Thus, the loss of sialic acid differentially affects the trafficking of B and T cells to liver, but not other organs.

Here we did not explore in great detail the fate of enzymatically desialylated lymphocytes, as such treatments can affect many types of sialylated glycans beyond O-glycans, and even glycolipid glycans. Others have reported that neuraminidase treatment of platelets led to their clearance by hepatic Kupffer cells, which was promoted by the Ashwell–Morell receptor (AMR) or asialoglycoprotein receptor[36,37]. Thus, it is possible that AMR and/or other unidentified related hepatic molecules may be involved in the active enrichment of desialylated T cells, but not B cells, in the liver. Interestingly, adhesion and/or migration of naïve subset-

specific lymphocytes have been reportedly regulated by the expression level of surface L-selectin, distinctive usage of chemokine receptors, and possibly other molecules[30,31,38–43]. Our results indicate that sialylation, as a novel mechanism, contributes to this process. Of note, intravascular staining has established that the vast majority of lymphocytes detected in liver and lung are located in the vasculature, but not in the interstitial tissues[44]. In our study, the livers and lungs were perfused with cold PBS, which will likely flush away most of the transferred cells in the precapillary vessels. The reduced B cells recovered from transfer experiments thus suggest that, in addition to its pivotal role in mediating lymphocyte to lymphoid tissues, sialic acid on B cells may be important for the retention of B cells in the capillary vessels of non-lymphoid organs.

Our results provide novel insights into the roles of B cell O-glycans, which have not been extensively studied previously. The

O-GalNAc O-glycans are critical in mediating neutrophil and activated T cell trafficking[5,45], as well as establishment and maintenance of T cell populations in the periphery[14]. Both $Tsyn^{-/-}$ and $C2GnT1^{-/-}$ neutrophils showed severely impaired recruitment into inflamed peritoneum, likely due to a defect in E-selectin binding[12,46]. By contrast, upon activation T cells quickly express enzymes generating O-glycans that terminate with the Sialyl Lewis X moiety, which allows the activated T cells to bind to selectins on endothelial cells and eventually promotes T cell extravasation into inflamed tissues[45,47]. However, there have been only a few studies on the potential roles of lymphocyte-expressed O-glycans in B cell biology. For example, similar percentage and numbers of resident B cells found in the lymphoid tissues of $C2GnT1^{-/-}$ and WT mice indicated that the absence of core-2 O-glycans on naïve lymphocytes does not affect B lymphocyte migration[12]. Decreased resident lymphocytes in certain PLNs of PPGALNAcT1-deficient mice were identified, but that seems attributable to reduced L-selectin ligand levels in the lymph node[13].

As global deletion of extended O-glycans causes embryonic death in mice, to achieve B cell-specific deletion of core-1 O-glycans, we crossed $Cosmc^{flox/flox}$ mice with mice expressing transgenic Cre-recombinase under the promoter of the Mb1 gene[21]. This approach efficiently generates BC-CosmcKO mice with a high rate of Cosmc deletion in B cells in the peripheral tissues (Supplementary Fig. 1). We observed a clear trafficking defect as was manifested in the substantially reduced percentage and numbers of resident B cells in lymph nodes and PPs, whereas B cells were mildly reduced in the spleen and bone marrow (Fig. 1). A disruption to homing via Cosmc deletion was confirmed by both short- and long-term adoptive transfer models. To our knowledge, in regard to single gene knockout models, Cosmc represents the only single gene identified to date that fully controls B cell trafficking to PLNs, GALTs, and non-lymphoid organs.

Previous studies have established the critical roles of lymphocyte-expressed L-selectin in initiating rolling[26,27], and β7 in forming firm attachment of lymphocytes in HEVs[28,29]. Lack of either molecule, or blockade of their interaction with receptors on endothelium, led to defective lymphocyte trafficking[48–50]. In addition, chemokine receptors have also been shown to be important to activate integrin[23,30–32]. When multiple chemokine receptors were desensitized, lymphocytes demonstrated impaired arrest and subsequently reduced homing[30]. We initially speculated that lack of extended O-glycans might lead to down-regulation of migration-related molecules. We observed, however, that Cosmc-deficient B cells, possibly through a compensatory mechanism, upregulated their surface expression of L-selectin, integrins, and chemokine receptors (Fig. 5d). This upregulation, however, does not functionally compensate for the deficiency caused by the loss of Cosmc during transmigration. It is interesting to note that as a consequence of Cosmc deletion, the Cosmc-deficient B cells may reprogram in a sense to overcome the extravasation blockade by upregulating L-selectin, which led to the increase of velocity of B cell during initial rolling. It is clear, however, that none of these potentially compensatory effects can salvage the B cell migration defect resulting from Cosmc deletion.

In terms of the mechanism for B cell trafficking into lymph nodes, our results demonstrate a key role for sialylated O-glycans in chemokine receptor activity. We recently noted that most chemokine receptors, and especially the subfamily of CC chemokine receptors (CCRs), have potential O-glycosylation sites in their extracellular N-terminus, and thus could be functionally impacted by the loss of extended O-glycans[51]. For CCR7, it was reported that sialylation is important for its functions in promoting CCL19 induced breast cancer cell growth[52]. In addition, the presence of N-glycans and sialic acid on T cell-expressed

CCR7 can modulate its receptor functions[33]. Our results suggest that sialylated O-glycans of CCR7 are required for its functional activity. It is known that CCR7 is O-glycosylated at several sites in its N-terminal surface domain[33,34]. The detailed structures of the O-glycans in B cell CCR7 of mouse or human origin remain unknown, but our results suggest that core 1-type O-glycans occur on murine CCR7 (Supplementary Fig. 7). However, we are cautious to interpret the contribution of potential Cosmc-dependent CCR7-ligand interaction to B cell homing because the presence of CCR7 appears dispensable for B cell migration[39]. But in addition, Cosmc-deficient B cells exhibit impaired response to CXCL12 (CXCR4 ligand), but unaltered response to CXCL13 (CXCR5 ligand). Thus, aspects of known chemokine signaling are clearly impaired in Cosmc-deficient B cells, but the overall functional chemokine signaling pathways required to mediate the activation of sequential integrins is poorly understood and remains to be explored in more detail.

Reflective of that, Cosmc-deficient B cells showed normal attachment to the HEV when measured by IVM. Thus, our results indicate that future studies should explore overall signaling responsiveness in terms of potential O-glycosylation on many migration-related molecules. One such case is human CD99, which is heavily O-glycosylated and has been shown to be pivotal in mediating the diapedesis of monocytes through endothelial junctions[53,54]. At present, however, the homolog of CD99 in mice has not been identified, but for human B cells it would be informative and worthwhile to investigate the functional contribution of O-glycans to CD99-mediated leukocyte transmigration. Moreover, PNA binding to immunoprecipitated CCR7 (Supplementary Fig. 7) also suggests a potentially important role of O-glycosylation in chemotactic responsiveness of lymphocytes, which contributes to the homing defects in Cosmc-deficient T cells that we observed in our recent study[14]. So far, no known glycan-binding protein (GBP) or lectin other than L-selectin expressed by lymphocytes has been reported to be important in lymphocyte homing to lymph nodes. Thus, future studies should consider the possibility that loss of sialylated O-glycans in Cosmc-deficient B cells might impair B cell interactions with an as yet unidentified adhesive or signaling GBP that recognizes normal O-glycans. In addition, recent notable technical advances have been made in visualizing how normal lymphocytes undergo transendothelial migration[55]. Such novel methodology may reveal more details about the behavioral changes of Cosmc-deficient B cells during their migration through HEVs.

In addition, our results indicate a complex regulation of B cell development by Cosmc. BC-CosmcKO mice showed dynamic changes in frequencies and absolute numbers of B lineage progenitors, which suggest that Cosmc is required for the normally progressive development of B cell in the bone marrow. The basis for the upregulation of surface expression of certain B cell subset markers is unclear and further studies are warranted to better understand the roles of Cosmc in regulating B cell development, localization, germinal center B cell response, and other functions. The increase of T cell subsets in the spleen of BC-CosmcKO mice is also interesting, which could be due to unidentified lymphocyte homeostasis mechanisms. Considering B cells play a role in T cell priming[22,56], it would be intriguing to examine how the Cosmc-deficient B cells interact with T cells under certain disease settings.

Our studies conclusively demonstrate that deletion of Cosmc in B cells alters their development and ablates their ability to migrate to lymph nodes. Particularly, Cosmc-deficient B cells are functional to roll and firmly attach to endothelium, but defective in their transmigration across the HEV barrier into lymph nodes. Future work is warranted to uncover the O-glycan-bearing molecules on B cells and their potential recognition partners within the endothelium. Here we reveal a novel glycosylation-based mechanism for

lymphocytes to access lymph nodes and provide new perspective for understanding lymphocyte trafficking in human health and disease.

## Methods

**Mice.** *Cosmc*[f/f] females were created from our previous work[18], and crossed with *Mb1*-Cre transgenic male mice (kindly provided by Dr. Michael Reth) to generate B cell-specific *Cosmc* knockout line. B cell-specific *Cosmc* knockout line were co-housed with WT littermate under specific pathogen-free conditions (21.7 ± 0.6 °C, 45 ± 10% humidity, and 12-h light cycle 6 am–6 pm) at Harvard Medical School in accordance with approved Institutional Animal Care and Use Committee (IACUC) protocols (Beth Israel Deaconess Medical Center, Harvard Medical School). All mice used in this study are male mice at 8 weeks old. All mice were euthanized by carbon dioxide overdose in a euthanasia chamber. Mouse genotypes were determined by PCR with primers for *Mb1*-Cre (hCre dir forward primer 5′-CCCTGTG GATGCCACCTC-3′, hCre reverse primer: 5′-GTCCTGGCATCTGTCAGAG-3′), and *Cosmc*[flox] (Forward primer: 5′-GCAACA CAAAGAAACCCTGGG-3′, Reverse primer: 5′-TCGTCTTTGTTAGGGGCTTGC-3′).

**B cell isolation, RT-PCR, and enzyme assays.** B cells from WT and BC-*Cosmc*KO mice were isolated using B cell isolation kit (Miltenyi Biotec, Cat#130-090-862) with purity over 92% as measured by CD19 positivity by flow cytometry.

*Cosmc* gene expression was measured by semi-quantitative RT-PCR. Briefly, total RNA isolated from WT and BC-*Cosmc*KO B cells using RNeasy Mini Kit (QIAGEN Ref#74104) was dissolved in RNase-free water. One microgram of total RNA from both groups was used for the synthesis of first strand cDNA using reverse transcriptase (SuperScript III, Invitrogen Ref#18080-044). PCR was performed with Phusion High-Fidelity PCR kit (New England Biolabs) in a 25 μl reaction system with primers (Forward primer 5′-ATCACTATGCTAGGCCACA TTAGGATTGGA-3′, Reverse primer 5′-GGAGGTAAGAAAACCAATGCAT CATTGAAAA-3′). B-actin was used as loading control (ACTB Forward primer: 5′-GGCTGATTCCCCTCCATCG-3′, Reverse primer: 5′-CCAGTTGGTAACA ATGCCATGT-3′). PCR products were analyzed by electrophoresis on a 1% Tris-acetate EDTA agarose gel.

For T-synthase and α-Mannosidase activity assays, isolated splenic B cells from both WT and BC-*Cosmc*KO mice were lysed in Tris-Buffered Saline containing 0.5% Triton X-100 and cOmplete-Mini protease inhibitor (Roche, Ref#11836170001) cocktail on ice. 10 μl cell extract supernatants were added to a final 50 μl reaction system, for T-synthase activity, containing 1000 μM GalNAc-α-4-(MU), 500 μM UDP-Gal, 20 mM MnCl$_2$, 0.2% Triton X-100, 800 units of *O*-glycosidase, in 50 mM MES-NaOH buffer (pH 6.8), or a 50 μl reaction system, for α-Mannosidase activity, containing 100 mM Man-α-4-(MU), 0.2% Triton X-100, 20 mM Tris-HCl (pH 7.8), for 45 min at 37 °C in a 96-well black plate. Reactions were stopped by adding 100 μL of 1.0 M glycine-NaOH (pH 10.0) and the relative fluorescence intensity were measured on a Victor Multiple-Label Counter (PerkinElmer) using umbelliferone mode.

**Antibodies and flow cytometry.** The antibodies were purchased from BD, Bio-legend, eBioscience, and listed as follows: CD19, B220, CD62L, β7, α4β7, CXCR5, CCR7, CXCR4, CD43, Ly51 CD24, CD23, CD21, IgM, IgD, Thy1.2-PE, or PerCp, or FITC, or PE-Cy7, or APC-Cy7, or Brilliant Violet 510, or Alexa Fluor-700, or Pacific Blue. Antibodies used in ELISA are from Southern Biotech or Thermo Scientific. Anti-Tn antibody prepared in the lab[57], was conjugated to Alexa Fluor 647 according to manufacturer's protocol (Thermo Scientific A20173). Biotinylated lectins were purchased from Vector Laboratories and incubated with final concentration at 2 μg per ml. Single-cell suspensions prepared from spleen, bone marrow from both femurs, blood, lymph nodes, Peyer's patches, and liver were stained with indicated antibodies at 1:100 dilution (except for α4β7-PE at 1:30 dilution) on ice (except for CCR7-FITC at room temperature) for 30 min, and run on BD Calibur, or LSR II, or Cytoflex. Data were analyzed with FlowJo software.

The Gating Strategies used for all flow cytometry experiments are shown in Supplementary Fig. 8 (for Figs. 1, 3a–c, 4, 5, Supplementary Fig. 2) and Supplementary Fig. 9 (for Figs. 2a, c, d, 4a, b, 5a–c, Supplementary Figs. 1A, B, 3A, B).

**ELISA.** Sera collected from WT and BC-*Cosmc*KO mice were titrated and added in duplicate into 96-well plates (Corning) precoated with polyvalent goat antibody against mouse immunoglobulins (IgM, IgG1, IgG2b, IgG2b, IgG3, IgA from Southern Biotech), and followed by HRP-conjugated goat anti-mouse IgG (Southern Biotech and Fisher Scientific) and then TMB ELISA substrate (Abcam). Absorbance was measured at 450 nm with a Multiskan Spectrum spectrophotometer (Thermo Scientific). The mouse immunoglobulin concentration was calculated from a curve constructed using mouse immunoglobulin standard as listed: IgM, IgG1, IgG2b, IgG2b, IgG3, IgA (Southern Biotech).

**Characterization of B cell glycans.** Approximately 5 million splenic B cells were purified, homogenized and extracted. The cells were next lysed and homogenized prior to incubation with DTT (1,4-dithiothreitol) and IAA (iodoacetamide) to denature the proteins. After dialysis to remove the DTT and IAA, the proteins were trypsinized (TPCK-treated trypsin) and the peptides were recovered and purified on C18 column. The purified peptides were treated with PNGaseF to remove the N-glycans. N-glycans and PNGaseF-treated peptides were recovered and purified on C18 column. The N-glycans were next permethylated and analyze by MALDI-TOF spectrometry.

O-glycans were removed from PNGaseF-treated peptides incubation with NaBH$_4$. Salts were removed for the preparation with a Dowex 50 W X8 column and co-evaporation with a methanol/acetic acid solution. O-glycans were then purified on a C18 column, permethylated and analyze by MALDI-TOF spectrometry.

MS data were acquired on a Bruker UltraFlex II MALDI-TOF Mass Spectrometer instrument. The reflective positive mode was used, and data were recorded between 100 and 6000 *m/z* for N-glycans, and between 0 and 4000 *m/z* for the O-glycans. For each MS profile, the aggregation of 20,000 laser shots or more were considered for data extraction. Only MS signals matching an N-/O-glycan composition were considered for further analysis. Subsequent MS post-data acquisition analysis was made using mMass[58,59].

**Characterization of IgG glycosylation.** IgG was purified from three WT and three BC-*Cosmc*KO mice sera using protein G agarose beads (Roche). Briefly, 50 μL protein G agarose beads were equilibrated with 2 × 500 μL PBS, followed by centrifugation at 2000 × *g* for 30 s and removal of the supernatant. 80 μL of PBS and 10 μL mouse serum were added to the beads and incubated on a shaker for 1 h at RT. The beads solution was transferred to empty top tips (Glygen), followed by 3 × 80 μL PBS washing steps and IgG elution with 60 μL 0.1 M glycine-HCl pH 2.7. The elution fraction was neutralized with 6 μL 1 M Tris-HCl pH 8.6.

For subsequent SDS-PAGE 22 μL 4x non-reducing Laemmli buffer were added for denaturation at 95 °C for 5 min. Each sample was loaded in two lanes (duplicate MS analysis) and the cut bands were used for in-gel trypsin digestion as described elsewhere[60]. Next day the supernatant was removed from the gel pieces and 50 μL 50% acetonitrile were added and incubated on the shaker for 10 min at RT. Both supernatants were combined and dried in a speed vac concentrator. The samples were taken up in 20 μL water and diluted 3x in 0.1% formic acid.

Two microliters of each sample were used for C18-reversed phase-liquid chromatography-mass spectrometry analysis (C18-RP-LC-MS/MS) using an Ultimate 3000 nano LC coupled to an Orbitrap Fusion Lumos mass spectrometer (both Thermo Fisher). Samples were loaded onto a C18 precolumn (C18 PepMap 100, 300 μm × 5 mm, 5 μm, 100 Å, Thermo Fisher Scientific) with 15 μL/min solvent A (0.1% FA in H$_2$O) for 3 min and separated on a C18 analytical column (picofrit 75 μm ID × 150 mm, 3 μm, New Objective) using a linear gradient of 2% to 45% solvent B (80% acetonitrile, 0.1% FA) over 39 min at 400 nL/min. The mass spectrometer was operated under following conditions: The ion source parameters were 2100 V spray voltage and 200 °C ion transfer tube temperature. MS scans were performed in the orbitrap at a resolution of 60000 within a scan range of *m/z* 400–*m/z* 1600, a RF lens of 30%, AGC target of 1e5 for a maximum injection time of 50 ms. The top 15 precursors were selected for MS$^2$ in a data dependent manner, within a mass range of *m/z* 550*m/z* 1600 and a minimum intensity threshold of 1e5 and an isolation width of 1.5 *m/z*. HCD was performed in stepped collision energy mode of 30% (±5%) and detected in the orbitrap with a resolution of 30000 with the first mass at m/z 120, an AGC target of 2e5 and a maximum injection of 250 ms.

EThcD spectra were acquired in a product ion-dependent manner ([HexNAc +H]$^+$-ion) based on the method above. Precursor isolation width was set to 1.2 *m/z*. Calibrated charge-dependent ETD parameters were used with supplemental activation collision energy of 25%, an AGC target of 2e5 and a maximum injection time of 250 ms.

Glycopeptide identification was performed using Byonic version 3.5 (Protein Metrics Inc.). Trypsin was set a protease with a maximum of two missed cleavage sites, the precursor and fragment mass tolerance was 10 ppm. The glycan database was "N-glycan 309 mammalian" and for O-glycans it contained the Tn antigen and non-, mono- and disialylated core 1 O-glycans. The following modifications were allowed: carbamidomethyl (Cys; fixed), oxidation (Met; variable, common 1), pyroglutamine on N-term (Gln; variable, rare 1), acetylation N-term (variable, rare 1), deamidation (Asn; variable, common 1), formylation N-term (variable, rare 1). Glycopeptides with a score above 250 were selected and further manually inspected.

Relative quantitation of all glycopeptides was performed in an automated manner[61,62]. The glycopeptide reference list contained all glycopeptides that were identified based on MS$^2$ fragmentation but also lower abundant glycopeptides based on their exact mass, corresponding retention time, isotopic pattern and biosynthetic related glycan composition. Relative intensities were determined in duplicates per mice and averaged, resulting in three data sets for both WT and BC-*Cosmc*KO. A two-tailed unpaired *t*-test was performed to compare neutral and sialylated N-glycan classes. The O-glycan distribution of the IgG2b hinge-region tryptic peptide LEPSGPISTINPCPPCK and the mis-cleaved peptide KLEPSGPISTINPCPPCK were averaged for both peptide species.

**Confocal microscopy.** Spleen, lymph nodes, and Peyer's patches were harvested from mice and frozen in OCT at −80 °C. The frozen tissues were cut at 6-μm thickness. Sections were air-dried, fixed with cold 1:1 methanol/acetone fixative for 10 min at −20 °C. After being rinsed 3 times with PBS containing 0.05% Tween 20, tissues were blocked with 10% goat serum for 2 h and then incubated with

anti-mouse CD19-PE and anti-Thy1.2-FITC overnight. The sections were then counterstained with Hoechst 33342 and mounted using ProLong gold reagent. Tile scanned images were acquired with a Zeiss LSM880 confocal microscope and analyzed by ImageJ (Fiji).

**Homing assays**. The in vivo homing assay was performed as described with modifications[63]. Single cell suspensions were prepared from the spleens of donor mice and labeled with CellTrace Violet according to manufacturer's protocol. Internal control wild-type splenocytes were labeled with CFSE. Donor splenocytes ($2 \times 10^7$) and equal numbers of internal control ($1–1.2 \times 10^7$) were intravenously co-injected into recipient mice in a volume of 300 μl of PBS. For some experiments, donor WT and BC-CosmcKO splenocytes and internal control were co-injected into recipients. An aliquot of the injection mixture was analyzed by flow cytometry for the injected ratio of Violet$^+$B220$^+$(or CD19$^+$) Tn$^+$, or Tn$^−$/CFSE$^+$B220$^+$ (or CD19$^+$) cells (Ri). After either 2 or 20 h of migration, single cell suspensions of blood and tissues were prepared and stained with antibodies, and the percentage of CellTrace-Violet and CFSE was determined by flow cytometry. The ratio of Violet$^+$ B220$^+$(or CD19$^+$) Tn$^+$, or Tn$^−$/CFSE$^+$B220$^+$ (or CD19$^+$) cells within individual organs or blood (Ro) was measured, and the results were presented as the ratio of Ro/Ri in each tissue.

**Chemotaxis assay**. The responsiveness of splenic B cells to chemokines was examined using 6.5-mm Transwell inserts with a 5-μm pore size (Corning). Fresh single cell suspension from spleen was prepared in complete RPMI1640 and incubated for 30 min at 37 °C, and resuspended in RPMI with 0.5 % BSA (~$10^7$ cells/ml). 100 μl suspension was placed to each insert in a well containing 580 μl solution of chemokines (R&D Systems) with indicated concentration. Migration was allowed for 4 h at 37 °C. Cells migrated to the lower chamber were collected, counted, and analyzed by flow cytometry.

**Intravital microscopy**. Intravital microscopy of the inguinal lymph nodes was done based on previous studies[12,64,65]. 6- to 12-week-old male C57BL/6J mice were anesthetized by intraperitoneal injection of 10 ml/kg saline containing xylazine (10 mg/kg) and ketamine (100 mg/kg). B cells from WT or BC-CosmcKO male mice were purified by magnetic-activated cell sorting (Miltenyi Biotec, Auburn, CA), labeled with calcein (Thermo Fischer, USA) and injected into the cannulated left femoral artery. The cells were visualized in the right inguinal lymph node using a IV-500 microscope (Mikron Instruments, Simi Valley, CA) equipped with an sCMOS camera (pco.edge 4.2, PCO-Tech Inc., Romulus, MI) and stroboscopic epifluorescence illumination. Data were recorded on a high speed video recorder (DVR Express® Core 2, IO Industries, Ontario, Canada) and analyzed as previously described[65,66]. Briefly, cells were considered noninteracting if they moved at velocities similar to that of red blood cells (RBCs), whereas rolling was defined as cells moving at detectably lower velocities. Sticking was defined as cells that become immobile for more than 30 seconds[63]. After recording, a FITC-dextran solution was injected to visualize the lymph node microvasculature and determine vessel structure and venular order[64,65]. Data were analyzed using two-way analysis of variance (ANOVA) with Bonferroni's multiple comparisons, and reported as mean ± SEM.

**Immunoprecipitation of CCR7 complex and western blotting**. CCR7, Abcam (Catalog number: ab32527), Protein A/G Magnetic beads, Thermo Scientific (Product Number: 88802), ECL Anti-mouse IgG HRP linked whole antibody from Sheep (GE Healthcare Cat#NA931V), biotinylated PNA (Vector Laboratories), and streptavidin (Vector Laboratories) were used in the experiment. The following dilutions of antibodies and lectins were used: CCR7 (1:5000), Biotinylated PNA (diluted to 1 μg/ml), streptavidin at 1:10,000 dilution in TTBS.

$10^7$ splenocytes pooled from three male B6 mice at 8 weeks old were lysed using 700μl of NP 40 lysis buffer (10 mM Tris-HCL, 150 mM NaCl, 2 mM EDTA, 1 mM DTT, and 1% NP 40) containing protease inhibitors. Cells were incubated on ice for 15 min and sonicated to break the cells followed by 15 min further incubation. Lysates were centrifuged at $13,000 \times g$ for 5 min at 4 °C. Supernatant were collected and aliquoted and stored at −80°C.

Two micrograms of anti-CCR7 or control antibody was absorbed on 25μL of 50% Protein A/G Magnetic Beads (Thermo Scientific) for ~2 h at 4 °C (25 rpm). Washed 2x with 1 ml of washing buffer (50 mM Tris-HCl, pH 7.4, 150 mM NaCl, 1.5 mM MgCl$_2$ and 0.2% Triton X-100) for 5 min each time. 125 μL of Lymphocyte extracts from wild-type mice was incubated with the preparation of the anti-CCR7 or control beads for 3 h (25 rpm) at 4 °C. Beads were pelleted and washed 3X with the washing buffer. The preparation was divided into two equal halves and one was processed for sialidase treatment as described in the company's protocol. The preparation was treated with 1x sample buffer (SDS-page) and boiled for 10 min and the eluted materials was aliquoted and the sample was loaded on SDS-PAGE system (one half for CCR7 and the other one for PNA staining).

Samples run on SDS-PAGE gels were transferred onto PVDF membrane by wet-transfer system. Membranes were blocked with 5% non-fat milk in 1x TBST (50 mM Tris, pH 7.4, 150 mM NaCl, 0.1% Tween-20) for 1 h at RT and further incubated with diluted primary antibody (overnight) with 1x TBST 5% non-fat milk. The membranes were washed 2x with 1x TBST for 5 min each followed by incubation with HRP-conjugated secondary antibody prepared in 1x TBST 5%

non-fat milk (1 h) and the membranes were washed 5x with 1x TBST and developed. Signals were detected by using ECL Prime Western Blotting Detection Reagent. For PNA, PVDF membrane was blocked with 5% BSA for 1 h. Biotinylated PNA was prepared in 1% BSA and secondary streptavidin reagents were prepared in 0.5% BSA.

**Statistics**. Unless stated otherwise, Group comparisons were analyzed using an unpaired two-tailed unpaired Student's t test with Prism software.

**Reporting summary**. Further information on research design is available in the Nature Research Reporting Summary linked to this article.

## Data availability
All data are available from the authors or are included in the supplementary data files. Source data are provided with this paper.

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

## Acknowledgements

The authors thank Dr. Michael Reth and Dr. John Manis for providing *Mb1*-Cre mice. We thank the BIDMC Histology, Confocal Microscopy, and Flow cytometry Core facility staff for their assistance. We thank Dr. Jamie Heimburg-Molinaro for help in manuscript editing and review; members of the Cummings lab and Mark B. Jones for helpful discussions. We thank Dr. Rodrigo J. Gonzalez for help in video editing. This work was supported by the HMS Center for Immune Imaging, and National Institute of Health Grant **U01CA168930** to T.J. and R.D.C.

## Author contributions

R.D.C. and J.Z. conceived and designed the project. J.Z. performed all experiments except for glycan release and analysis, which was conducted by S.L., IgG glycopeptide analysis by K.S., chemokine receptor biochemical analysis by R.P.A., and IVM by M.E. and U.H.v.A. J.W., Y.W., and T.J. generated the floxed *Cosmc* mice. J.Z., K.S., S.L., M.E., and R.P.A. analyzed and interpreted the data with the assistance of M.R.K. R.D.C., R.P.A., and J.Z. wrote the manuscript, which was edited and approved by all authors.

## Competing interests

The authors declare no competing interests.
