## [Peer Review File · Nature Communications]

Reviewers' comments:

Reviewer #1 (B cell biology, microscopy) (Remarks to the Author):

The manuscript describes a role for the T-synthase folding chaperone Cosmc in regulating B cell homing, in particular the entry of B cells into secondary lymphoid organs. While this observation is interesting, the manuscript fails to provide mechanistic insights beyond this initial observation and lacks stringency, as well as a proper discussion of the findings. For now, it seems somehow unfinished. Major concerns:

The authors do not observe differences in rolling/adhesion behavior, but reduced accumulation of B cells in SLOs, so they conclude that Cosmc affects the later phases of endothelial transmigration. This should be analyzed more in detail, and molecular mechanisms should be analyzed or at the very least discussed.

A more detailed analysis of the Cosmc deficient mice would be required, and data on homing behavior of the main hematopoietic subsets should be included. The paper focuses on the homing effect on B cells, but other important functions of are not mentioned, it is not clear whether they were just not analyzed. For example, in addition to homing, glycosylation plays an important role for other processes related to B cell biology, such as antibody glycosylation. It should be tested whether Cosmc is involved in those processes, i.e. is antibody glycosylation changed in the Cosmic knockout mice? The glycosylation pattern of antibodies crucially determines their function, and O glycosylation defects have been associated with autoimmunity. In that respect it would be interesting to know whether the Cosmc deficient mice develop autoimmunity with age.

The authors find that B cells have a homing defect and are reduced in the spleen, but on the other hand the spleen weight is increased. This seems contradictory and should be explained, and other main immune cell subsets in the spleen should be analyzed and compared to wt, and the results should be discussed. In general, the manuscript would benefit from a more elaborate discussion. Another example is in Line 165, where higher expression levels of chemokine receptors were observed, which seems counterintuitive given the homing defect. Again, this is not at all discussed.

Minor issues:

Line 106: which background factors?

Line 115: unclear to me, what the previous study is? what is consistent?

Line 126: It may make more sense to move this paragraph upstream

Line 134: Why do they look for Thy1.2. in particular? Why do they look in the liver?

Line 143: Why are there no movies supplied?

Line 146: How did they determine the sticking and rolling fraction?

Extended Data Fig. 1a – please invert for better visibility

Fig. 1b/c description missing (WT/KO)!

Had problems finding E

Fig. 1 G in the text it says there is CD19 staining but it's only written B cells in the legend. Scale bar is missing

Fig. 2 – are the two time points really sufficient? How about (immunizing and) looking at later timepoints in the BM?

Fig. 3 a+b legend explaining symbols of glycosylation would be helpful

For Extended data figure 4 D very low cell numbers, it would need more cells to support that statement

Reviewer #2 (Glycomics, mass-spec analysis) (Remarks to the Author):

The manuscript by Cummings and co-workers convincingly shows that B-cell mucin-type elongated O-glycosylation is important for B cell homing to peripheral lymphoid organs. The manuscript point to the fact that the glycosylation of chemokine receptors may play an important role in B cell homing. The findings are of major significance as they shed light on the function and organization of immunological responses. I have a few comments which the authors may want to consider:

While the study focuses on B cell homing to peripheral organs, I wonder how the COSMC mutation affects B cell development, abundance, lifetime, clearance, as well as B cell function in a broader sense.

Also, the data pointing towards the involvement of the chemokine axis in causing O-glycosylation dependent B cell homing appear not fully conclusive, and further experiments may be needed to substantiate this.

Reviewer #3 (Chemotaxis, B cell migration) (Remarks to the Author):

In the present manuscript Zeng et al. investigate the phenotype of a novel mouse that carries B cell-restricted deletion of the *Cosmc* gene. The authors report that B cells show a different pattern in O-glycosylation in B cells and that homing of these cells is strongly impaired to lymphoid and non-lymphoid organs. Mechanistically they show that the different interaction steps of homing B cells to HEVs are not affected. Instead they provide some preliminary evidence that impaired glycosylation of chemokine receptors might be responsible for impaired homing. This study is of clear interest but the current version is rather preliminary.

Major points:

- 1) The authors postulate that impaired homing of B cells is due to impaired glycosylation of chemokine receptors, in particular CXCR4 and CCR7. However only circumstantial evidence has been provided that this is the case. They therefore must show that the glycosylation pattern of these chemokine receptors is actually different in *Cosmc*-deficient B cells but e.g. not in T cells. Further, they must show that either binding of the corresponding chemokine(s) to their receptor is affected or subsequent receptor signaling is impaired – or both!
- 2) The authors should also create mice with specific deletion of *Cosmc* in T cells and DCs and also characterize homing of these immune cell subsets: This is important since it has been shown that CCR7 is indispensable for homing of T cells and DCs while CXCR5 is more important for B cell homing. However, in the present study the authors demonstrate that *Cosmc*^{-/-} B cells show unimpaired migration to CXCL13, the ligand for CXCR5.
- 3) The authors describe a 50% reduction in splenic B cells, but also report splenomegaly in *Cosmc*-deficient mice. Which cells contribute to the observed splenomegaly? Furthermore does *Cosmc*-deficiency in B cells affect the architecture of the splenic white pulp? (GC-formation, follicle formation, MZ, marginal sinus, T cell positioning?)
- 4) The authors report that the N-glycosylation pattern is not affected in *Cosmc*-deficient B cells. However, at least to this reviewer, ext. figs 3a and 3b show different patterns of glycosylation.
- 5) The authors report a 50% reduction of B cells in the bone marrow. This has to be further investigated. Is B cell maturation in the bone marrow affected, or recirculation of mature B cells?
- 6) Plasma cells have been shown to take advantage of CXCR4. Is plasma cell/blast homing to bone marrow affected in the mice analyzed?
- 7) Fig 1J indicates that T cells are missing in PP in *Cosmc*-deficient mice. Quantitative analysis is needed to address whether this is indeed the case. Also, the section plane used for PP analysis is unusual. Sections showing FAE in the dome region as well as mucosa and serosa should be provided

Minor:

- 1) Give reference for the statement the the TN antigen is abnormally expressed
- 2) Give reference for the Cosmc fl/fl mouse used in this study
- 3) More information about COSMC must be provided in the introduction
- 4) The authors frequently refer to reduced cell numbers as 'fold reduction'. This phrase is scientifically incorrect. Therefore use e.g. 'reduced to 5%' instead of '20-fold reduction'

Response to Reviewers

REVIEWER 1 COMMENTS

COMMENTS FOR THE AUTHOR: The manuscript describes a role for the T-synthase folding chaperone Cosmc in regulating B cell homing, in particular the entry of B cells into secondary lymphoid organs. While this observation is interesting, the manuscript fails to provide mechanistic insights beyond this initial observation and lacks stringency, as well as a proper discussion of the findings. For now, it seems somehow unfinished.

Author Response: We thank the reviewer for appreciating the importance of our work. We agree with the reviewer that the mechanism of B cell diapedesis into secondary lymphoid tissues remains unclear. However, we would like to point out that, to the best of our knowledge, *Cosmc* is the first gene that has been identified to be essential in mediating B cell diapedesis, but not in the early steps. Our study thus ushers in a new direction for dissecting B cell transmigration. Additionally, we have examined B cell development and immunoglobulin levels in the serum of BC-*Cosmc*KO mice. We included this set of data in a new **Fig. 2**. In this revised manuscript, we have provided a more detailed discussion about our findings. Overall, our results conclusively demonstrated that *Cosmc* determines B cell migration.

We have addressed the reviewer's concerns point-by-point as follows:

1. *The authors do not observe differences in rolling/adhesion behavior, but reduced accumulation of B cells in SLOs, so they conclude that Cosmc affects the later phases of endothelial transmigration. This should be analyzed more in detail, and molecular mechanisms should be analyzed or at the very least discussed.*

Author Response: We appreciate the reviewer's point and we agree a more detailed analysis on *Cosmc*-deficient B cell transmigration study through endothelium would be meaningful towards understanding the molecular mechanism of transmigration. However, as there is no appropriate model system to study lymphocyte transmigration (diapedesis) into the lymph node and other technical limitations such a study is not possible at this point. We no doubt that in future studies after our findings are published, that this whole area of research may be opened. We also appreciate the reviewer for asking us to discuss more about B cell transmigration; as suggested, we have added a discussion section with our deeper perspective about the possible mechanisms of how *Cosmc* affect B cell transmigration in the revised version (pages 11,12).

2. *A more detailed analysis of the Cosmc deficient mice would be required, and data on homing behavior of the main hematopoietic subsets should be included. The paper focuses on the homing effect on B cells, but other important functions of are not mentioned, it is not clear whether they were just not analyzed. For example, in*

addition to homing, glycosylation plays an important role for other processes related to B cell biology, such as antibody glycosylation. It should be tested whether Cosmc is involved in those processes, i.e. is antibody glycosylation changed in the Cosmic knockout mice? The glycosylation pattern of antibodies crucially determines their function, and O glycosylation defects have been associated with autoimmunity. In that respect it would be interesting to know whether the Cosmc deficient mice develop autoimmunity with age.

Author Response: We appreciate the reviewer for asking about the detailed analysis of the BC-CosmcKO mice, as we agree that the detailed characterization of the mice that we have generated is important. Our work on global *Cosmc* KO in mice is embryonic lethal and we have already published the work (ref. 18, Wang *et al. PNAS*, 2010). In addition, we also have generated T cell specific *Cosmc* KO mice, and this work was published late last year, which shows *Cosmc* impacts the behavior of T cell migration (ref. 14, Cutler *et al. Glycobiology*, 2019). In this manuscript we focus on dissecting the role of *Cosmc* in B cell homing, and we have documented that *Cosmc*-deficient B cells shows a profound and distinct migration defect. We appreciate the reviewer's suggestion to analyze other aspects of B cell biology, we agree the importance of such analysis and accordingly we have updated the current manuscript. We found that *Cosmc* also plays critical roles in other aspects of B cell biology, and we have added a new figure (**Fig. 2**) showing that the development of B cells was disrupted in the bone marrow and in the spleen of B cell-specific *Cosmc* knockout mice. The migration defect of *Cosmc*-deficient B cell is so unique and pivotal that we believe that this finding should be the focus of this study.

We appreciate the reviewer for pointing out the importance of glycosylation of antibodies, which is critical to many aspects of B cell biology/function and *Cosmc* may have important role in this process. Although O-glycosylated human Immunoglobulins has been well-documented, only one study reported O-glycosylation of the hinge region of IgG2b in mouse. In the revised version of the manuscript, we have explored this unusual feature and confirmed that murine IgG2b is O-glycosylated. Purified IgG2b from WT mice carry mono- and disialylated core 1 O-glycans, while the BC-CosmcKO IgG2b contains exclusively the Tn antigen on IgG2b hinge-region. This is exciting new information and we are happy the reviewer felt this was important to add. We have added this new set of data as **Fig. 4C** and **Supplementary Fig. 5**. But just as importantly, we also show that the loss of *Cosmc* does not alter the N-glycosylation pattern of IgG from BC-CosmcKO mice, as shown in **Supplementary Fig. 4**.

We appreciate the reviewer's suggestion to explore the potential development of autoimmunity in BC-*Cosmc*KO mice. This is a large issue that is really secondary to the main findings of our paper under consideration. As the reviewer would expect, we are currently investigating these mice and the results will be reported, separately.

3. *The authors find that B cells have a homing defect and are reduced in the spleen, but on the other hand the spleen weight is increased. This seems contradictory and should be explained, and other main immune cell subsets in the spleen should be analyzed and compared to wt, and the results should be discussed. In general, the manuscript would benefit from a more elaborate discussion. Another example is in Line 165, where higher expression levels of chemokine receptors were observed, which seems counterintuitive given the homing defect. Again, this is not at all discussed.*

Author Response: We appreciate the reviewer's comment. We have analyzed the T lymphocyte subsets in peripheral lymphoid tissues and this new data has been integrated as **Supplementary Fig. 1D**. The data suggests that the increased size of spleen was largely due to the presence of significantly more T cells, both CD4 and CD8 T cells, in the spleen. We have provided our insights in the discussion regarding the splenic enlargement and the increased surface expression level of chemokine receptors.

4. *Line 106: which background factors?*

Author Response: We thank the reviewer for pointing out the confusing part in our text, and we apologize for the confusion. *Cosmc*-deficient B cells show similar defective trafficking pattern in both WT and BC-*Cosmc*KO recipient mice, indicating it is a *cell-intrinsic* phenotype. We have rephrased our expression and replaced with "in a cell-intrinsic manner" in the text (top of page 6).

5. *Line 115: unclear to me, what the previous study is? what is consistent?*

Author Response: The previous study (ref. 25, Comelli, EM., *et al. J Immunol* 2006, page 6) showed the presence of various N-glycans on murine B cells. We have performed the N-glycan profiling of both WT and *Cosmc*-deficient B cells and showed similar pattern to what was observed in that study.

6. *Line 126: It may make more sense to move this paragraph upstream*

Author Response: N-glycan profiling of murine B cells has been reported. However, analysis of O-glycans on primary murine B cells by mass spectrometry has not been studied. We believe that it would be more logical to start with N-glycan then followed by O-glycan profiling (pages 6-7).

7. *Line 134: Why do they look for Thy1.2. in particular? Why do they look in the liver?*

Author Response: We thank the reviewer for pointing out the confusion, and we agree that the reason for using Thy1.2 and the liver in this case was not clearly written. Thus, we have updated the introduction in the revised version of this manuscript. We used Thy1.2 simply because it is a widely used pan-T cell marker. We also used this marker in the immunofluorescence assay to show that, in general, the B cell follicle appears normal in spleen of BC-CosmcKO mice. On the other hand, as explained in the introduction, previous studies by Gesner and Ginsberg (ref. 8) have shown that after enzymatic removal of sialic acids, lymphocytes accumulated in the liver. However, curiously this phenotype has not been reported in any of the sialyltransferase gene knockout mice. So, in our study, we wanted to analyze the lymphocytes in a greater detail which are enriched in the liver as well as other organs.

8. *Line 143: Why are there no movies supplied?*

Author Response: We have attached 2 representative movies in supplementary information.

9. *Line 146: How did they determine the sticking and rolling fraction?*

Author Response: Cells were considered noninteracting if they moved at velocities similar to that of red blood cells (RBCs), whereas rolling was defined as cells moving at detectably lower velocities. Sticking was defined as cells that become immobile for more than 30 seconds. For more details, please see the study by Warnock RA., et al. JEM 1998 (ref. 59). These details have been added to the Methods section.

10. *Extended Data Fig. 1a – please invert for better visibility*

Author Response: We thank the reviewer for pointing this out, but inversion of the panel seems not to improve the visibility, so we have added an arrow to point out the *Cosmc* DNA bands in B cells from both WT and BC-CosmcKO mice in the revised version of the manuscript.

11. *Fig. 1b/c description missing (WT/KO)!*

Author Response: We thank the reviewer for pointing this out, and we have added the description in the revised version of the manuscript, and above the graph in **Fig. 1a** to denote the WT and KO color scheme for the bar graphs.

12. *Had problems finding E*

Author Response: We thank the reviewer for pointing this out, and we have adjusted the panel E in the revised version of **Fig. 1** of the manuscript.

13. *Fig. 1 G in the text it says there is CD19 staining but it's only written B cells in the legend. Scale bar is missing*

Author Response: We thank the reviewer for pointing this out, and we have added CD19 to the legend. We also have added the scale bar for each picture in the revised version of the manuscript.

14. *Fig. 2 – are the two time points really sufficient? How about (immunizing and) looking at later timepoints in the BM?*

Author Response: We appreciate the reviewer's suggestion and we agree that many time points would be ideal. In our study, however, we just used two time points which we feel is appropriate and based on other published work in which two time points are widely used for adoptive transfer. Thus, we believe these two time points are sufficient to answer the question. Intravenously injected lymphocytes migrate to lymph nodes within seconds and we have used two different time points which allow sufficient time for cells to travel to the lymph nodes.

We have intraperitoneally immunized both WT and BC-*Cosm*CKO mice with 100 µg nitrophenyl conjugated to keyhole limpet hemocyanin (NP-KLH) in complete Freund adjuvant (CFA). We then examined the B220^{lo} CD138^{hi} plasma cells at day 7. Our initial analysis showed comparable percentage of plasma population in the bone marrow of both WT and WT and BC-*Cosm*CKO mice. Our focus in this study is on naïve B cells. We believe that it would be more suitable in a future separate study to address how *Cosmc* regulates activated B cells, which we feel is beyond the scope of the homing issues in our report.

15. *Fig. 3 a+b legend explaining symbols of glycosylation would be helpful*

Author Response: Yes, we thank the reviewer for pointing this out. We have added the symbol and depictions to the figure (now **Figure 4**) as suggested in the revised version of the manuscript. Similarly, we added the symbol to **Supplementary Figure 3**.

16. *For Extended data figure 4 D very low cell numbers, it would need more cells to support that statement.*

Author Response: The very low cell numbers shown in orange square is one of the important phenotypes that we identified and is consistent with what Gesner and Ginsberg observed. Those are splenocytes treated with neuraminidase, which resulted in the substantially reduced homing to peripheral lymph nodes. As a comparison, as shown in the right side of **Extended data Figure 4D**, when splenocytes treated with PBS, or untreated as internal control, both B and T cells within the transferred population showed normal migration to peripheral lymph

nodes. As stated in the methods, at least 1 million events, or the whole tissue was prepared, and all events were collected.

REVIEWER 2 COMMENTS

The manuscript by Cummings and co-workers convincingly shows that B-cell mucin-type elongated O-glycosylation is important for B cell homing to peripheral lymphoid organs. The manuscript point to the fact that the glycosylation of chemokine receptors may play an important role in B cell homing. The findings are of major significance as they shed light on the function and organization of immunological responses. I have a few comments which the authors may want to consider:

While the study focuses on B cell homing to peripheral organs, I wonder how the COSMC mutation affects B cell development, abundance, lifetime, clearance, as well as B cell function in a broader sense.

Author Response: We appreciate the reviewer's comments that the findings are of major significance. We likewise wonder how B cell specific *Cosmc* deletion impacts other aspects of B cell biology and accordingly have examined such impact on B cell development as well as immunoglobulin production. In this revised version of the manuscript, we have incorporated new data, as **Fig. 2**, which showed that the *Cosmc* plays an important role in regulating B cell development and antibody levels in the serum. We thank the reviewer for the suggestion.

Also, the data pointing towards the involvement of the chemokine axis in causing O-glycosylation dependent B cell homing appear not fully conclusive, and further experiments may be needed to substantiate this.

Author Response: We appreciate the reviewer's comments. Please see our detailed discussion that we added in the main text of this revised version of this manuscript highlighting the importance of O-glycans in B cell homing. Briefly, we conclusively demonstrate in our report that B cell-intrinsic O-glycosylation controls B cell homing. We also observed that *Cosmc*-deficient B cells showed impaired responsiveness to chemokines. However, the contribution of such responsive impairments to the *Cosmc*-deficient B cell homing is still debatable. Our intravital microscopy data showed that *Cosmc*-deficient B cells are still able to firmly attach to the HEVs, suggesting that the impaired chemotactic responsiveness somewhat is rescued by other signals. Nonetheless, as we pointed out in the main text, other migration-related molecules are also likely to be O-glycosylated, as so many glycoproteins are now expected to acquire O-glycans, but the repertoire of O-glycosylated glycoproteins in murine B cells is unknown at present and would require a major study. Our study in fact would be a starting point for future investigation to elucidate the remarkable effect of O-glycosylation on B cell homing.

REVIEWER 3 COMMENTS

In the present manuscript Zeng et al. investigate the phenotype of a novel mouse that carries B cell-restricted deletion of the Cosmc gene. The authors report that B cells show a different pattern in O-glycosylation in B cells and that homing of these cells is strongly impaired to lymphoid and non-lymphoid organs. Mechanistically they show that the different interaction steps of homing B cells to HEVs are not affected. Instead they provide some preliminary evidence that impaired glycosylation of chemokine receptors might be responsible for impaired homing. This study is of clear interest but the current version is rather preliminary.

Author Response: We thank the reviewer for appreciating the importance of our work. We agree with the reviewer that defining the molecular mechanism of B cell homing is important and our current manuscript in this direction is just the beginning. We appreciate the reviewer's suggestion. However, currently, we have the limitation of the available model systems and tools, to dissect B cell diapedesis into tissues, as at present there is no appropriate model system to study lymphocyte transmigration (diapedesis) into the lymph node. We believe that our discovery provides a novel perspective on the role of O-glycans on B cells and their role in transmigration of B cells to the tissues. Importantly, we provide evidence that the *Cosmc*-deficient B cells would now be a valuable tool to explore further towards this direction.

1. *The authors postulate that impaired homing of B cells is due to impaired glycosylation of chemokine receptors, in particular CXCR4 and CCR7. However only circumstantial evidence has been provided that this is the case. They therefore must show that the glycosylation pattern of these chemokine receptors is actually different in Cosmc-deficient B cells but e.g. not in T cells. Further, they must show that either binding of the corresponding chemokine(s) to their receptor is affected or subsequent receptor signaling is impaired – or both!*

Author Response: We thank the reviewer for the comments. As we stated above, due to format limitations, in our original manuscript, we were not able to articulate our insights into the role of chemotactically responsive impairments of *Cosmc*-deficient B cells in B cell homing defects. Please see our revised discussion in the main text. We have conclusively demonstrated that deletion of *Cosmc* in B cells leads to defective B cell homing. On the other hand, from *in vitro* trans-well migration assays, the chemotactic responsiveness of CCR7 severely, and CXCR4, to a lesser extent, was impaired in cells lacking *Cosmc*. However, because CCR7-KO B cells are still able to

migrate to peripheral lymph nodes, thus it is still insufficient to hypothesize that the *Cosmc*-deficient B cell homing defects are completely attributable to the impaired chemokine responsiveness of CCR7 or CCR7 related O-glycosylation. Our broader exploration of potential contributions of *Cosmc* to chemokine receptors is certainly intriguing and worthwhile to explore further because, as the reviewer pointed out, CCR7 is critical for T cells and dendritic cells migration. In this regard we conducted several new studies for this revision, and now we have provided one line of evidence indicating the presence of O-glycans on CCR7 (**Supplementary Fig. 7**). Another line of evidence came from the extensively O-glycosylated human CD99, which is already documented. But to conclusively demonstrate the O-glycans on CCR7, as well as other migration-related molecules, new methodologies would first need to be developed, which would be beyond the scope of current study, analysis of such O-glycosylation and potentially tyrosine sulfated peptide fragments will require synthetic standards to be definitive. We feel that our study suggests a new justification for future studies in the roles of O-glycans in B cell migration.

2. *The authors should also create mice with specific deletion of Cosmc in T cells and DCs and also characterize homing of these immune cell subsets: This is important since it has been shown that CCR7 is indispensable for homing of T cells and DCs while CXCR5 is more important for B cell homing. However, in the present study the authors demonstrate that Cosmc^{-/-} B cells show unimpaired migration to CXCL13, the ligand for CXCR5.*

Author Response: We agree with the suggestions of the reviewer. Considering the importance of CCR7 in guiding dendritic cells and T cells migration, creating mice with cell-specific deletion of *Cosmc* and studying their homing to tissues would provide deeper and broader understanding of the mechanistic roles of *Cosmc* in immune cell homing. In our recently published work (ref. 14, Cutler *et al*, *Glycobiology*, 2019), we have created T cell specific *Cosmc* deletion mice and analyzed T cell homing, which indicated impaired migration of *Cosmc*-deficient T cells to peripheral lymph nodes. Certain levels of similarity between the homing pattern of *Cosmc*-deficient T cells and that of CCR7-deficient T cells has been observed. However, we did not pursue the potential role of *Cosmc* in regulating chemokine responsiveness in *Cosmc*-deficient T cells. We will certainly continue to investigate further in that direction. As we mentioned above, a large body of work would be warranted to identify and investigate the contribution of *Cosmc* and O-glycosylation in general to the function of migration-related molecules, which we trust the reviewer will agree is beyond the scope of current study.

3. *The authors describe a 50% reduction in splenic B cells, but also report splenomegaly in Cosmc-deficient mice. Which cells contribute to the observed splenomegaly? Furthermore does Cosmc-deficiency in B cells affect the architecture of the splenic white pulp? (GC-formation, follicle formation, MZ, marginal sinus, T cell positioning?)*

Author Response: We thank the reviewer for the comments. The observed spleen enlargement in BC-*Cosmc*KO mice is due to the presence of significantly more T cells

in the spleen. We have analyzed the T cell composition in peripheral lymphoid organs and added as in **Supplementary Fig. 1D**. As the immunostaining in **Fig. 1** showed, the lymphoid follicles in BC-CosmcKO mice appear to be normal. We agree with the reviewer that further studies would be preferred, although we trust the reviewer will agree that this is beyond the scope of current study, i.e. to investigate how *Cosmc* regulates B cell activation, localization, and other functions. We will report those studies separately.

4. *The authors report that the N-glycosylation pattern is not affected in Cosmc-deficient B cells. However, at least to this reviewer, ext. figs 3a and 3b show different patterns of glycosylation.*

Author Response: We thank the reviewer's comments. To convincingly demonstrate that lack of *Cosmc* does not affect the general N-glycosylation of murine B cells, we developed an analytical method to examine the detailed N-glycans on all IgG isotypes from BC-CosmcKO mice. We have included those new data as **Supplementary Fig. 4** and we can conclude that deletion of *Cosmc* does not affect N-glycosylation on B cells.

5. *The authors report a 50% reduction of B cells in the bone marrow. This has to be further investigated. Is B cell maturation in the bone marrow affected, or recirculation of mature B cells?*

Author Response: We thank the reviewer for the suggestion to further investigate B cell maturation in the bone marrow or recirculation of the B cells. As suggested, we analyzed the B cell development in the bone marrow and found that the loss of *Cosmc* caused dynamic changes in B cell populations in the bone marrow. We have incorporated the data as a new figure (**Fig. 2**) in the revised version of the manuscript. The magnitude of reduction of *Cosmc*-deficient mature B cells in the bone marrow is far greater than those in the spleen suggested that the recirculation of splenic mature B cells to the bone marrow is possibly impaired. Although it could be also due to the dampened local maturation in the bone marrow of BC-CosmcKO mice, or both combined.

6. *Plasma cells have been shown to take advantage of CXCR4. Is plasma cell/blast homing to bone marrow affected in the mice analyzed?*

Author Response: We thank the reviewer for this suggestion. We reasoned that after immunization, the B220^{lo} CD138^{hi} plasma cell population in the bone marrow of both WT and BC-CosmcKO mice would be a good indication to answer whether the homing of plasma cells to the bone marrow would be affected by the loss of *Cosmc* or not. As we stated above, we have performed the immunizations, and our initial analysis showed comparable percentage of plasma population in the bone marrow of both WT and BC-CosmcKO mice, suggesting that the homing of plasma cells is not affected by *Cosmc*.

7. *Fig 1J indicates that T cells are missing in PP in Cosmc-deficient mice. Quantitative analysis is needed to address whether this is indeed the case. Also, the section plane used for PP analysis is unusual. Sections showing FAE in the dome region as well as mucosa and serosa should be provided.*

Author Response: We thank the reviewer for their critical reading. As suggested, we have analyzed the T cell composition of peripheral lymphoid organs, including PPs, and this new data is added as **Supplementary Fig. 1D**. The T cell percentage increased in PPs of BC-*Cosm*CKO mice. But the absolute number of T cell numbers reduced due to the rudimentary nature of PPs. We have updated with images of PPs from both WT and BC-*Cosm*CKO mice with complete structure.

Minor points:

- 1) *Give reference for the statement the the TN antigen is abnormally expressed*

Author Response: We thank the reviewer for the comment. We have provided more background about the Tn antigen in the introduction and related references.

- 2) *Give reference for the Cosmc fl/fl mouse used in this study.*

Author Response: We thank the reviewer for the comment. We have added the reference in the revised version of the manuscript.

- 3) *More information about COSMC must be provided in the introduction*

Author Response: We apologize for the lack of information regarding *Cosmc* in the introduction due to the format limitation. Thank you for pointing this out and we have updated the information in the introduction of the revised version of the manuscript and note the relevant references where previous *Cosmc* work was performed.

- 4) *The authors frequently refer to reduced cell numbers as 'fold reduction'. This phrase is scientifically incorrect. Therefore use e.g. 'reduced to 5%' instead of '20-fold reduction'*

Author Response: Thank you very much for the suggestion, and we have made changes in the revised version of the manuscript accordingly.

REVIEWERS' COMMENTS:

Reviewer #1 (Remarks to the Author):

In the present manuscript by Zeng et al., a novel role for the T-synthase chaperone Cosmc in B cell development and homing is demonstrated.

The revision of the manuscript has led to a clear improvement in quality. The authors have added experiments on the characterization of the B cell compartment in mice with a B cell-specific Cosmc deletion, along with other additional experiments, which altogether make the work more complete.

I only have a few comments:

1. The authors state that Cosmc mainly affects B cell homing to lymph nodes. However, in Fig. 2 of the revised paper they clearly demonstrate that also B cell development in the bone marrow is affected by Cosmc deletion. This should be emphasized and carefully considered in the interpretation/discussion, as it may also add to the homing effect observed. Along that line, the authors should consider choosing a different title. The current title doesn't reflect important findings shown in Fig. 2.

2. What is the time frame of the intravital movies? Which time span was recorded? Can the authors add a time stamp to the movies?

Along that line, what is the size of the field of view in the intravital movies? The authors should add some kind of reference, such as a scale bar. A movie legend is missing and should be added.

Typos:

L 94 B cell not B cells

L 95 block instead of blocks

Reviewer #2 (Remarks to the Author):

The authors have followed the suggestions and significantly improved the manuscript.

Reviewer #3 (Remarks to the Author):

After having carefully studied the revised manuscript I come to the following conclusions:

1) Contrary to common practice, the authors do not provide a manuscript in which the revised passages are marked in change-track mode. It is therefore rather difficult to identify those changes that have been made to the text and to figures.

2) One of my key points when reviewing the initial manuscript was that the study was rather preliminary and that further investigations should be done. In particular I suggested investigating into more detail the mechanisms underlying impaired homing of B cells via HEVs. Unfortunately this was not done. In their rebuttal letter, the authors write that such studies are currently not possible and that new experimental procedures need to be established. This is very surprising, however, as John Kehrl's group used multiphoton microscopy in a manuscript recently published in *iScience* to study in detail homing of lymphocytes via HEV (doi: 10.1016/j.isci.2019.05.040). That study actually provides a blueprint of the experiments needed here to gain further insights at what stage of their journey through HEV Cosmic-deficient B cells get stuck and how their behavior differs from that of their wild

type counterpart.

3) Was ref 14 (Cutler et al) already mentioned in the first version of the manuscript? I do not remember that the authors had adequately addressed impaired homing of *cosmc*^{-/-} T cells there. Anyway! In Cutler et, it was shown that T cell-specific deletion of *cosmic* leads to impaired homing of T cells. In the present manuscript analogous results are shown for B cells without appropriately elucidating the underlying mechanisms. The authors speculate that the function of chemokine receptors may be impaired. However, only transwell assays were performed but no further analysis is shown.

In summary, I maintain my opinion that the B cell-specific *cosmc*^{-/-} mice show an interesting homing phenotype, but the results presented here are too preliminary for possible publication in Nature Communications.

Response to Reviewers

REVIEWER 1 COMMENTS

COMMENTS FOR THE AUTHOR: In the present manuscript by Zeng et al., a novel role for the T-synthase chaperone Cosmc in B cell development and homing is demonstrated. The revision of the manuscript has led to a clear improvement in quality. The authors have added experiments on the characterization of the B cell compartment in mice with a B cell-specific Cosmc deletion, along with other additional experiments, which altogether make the work more complete.

Author Response: We appreciate the positive thoughts from this reviewer on our revised version of the manuscript.

I only have a few comments:

1. The authors state that Cosmc mainly affects B cell homing to lymph nodes. However, in Fig. 2 of the revised paper they clearly demonstrate that also B cell development in the bone marrow is affected by Cosmc deletion. This should be emphasized and carefully considered in the interpretation/discussion, as it may also add to the homing effect observed. Along that line, the authors should consider choosing a different title. The current title doesn't reflect important findings shown in Fig. 2.

Author Response: We appreciate the reviewer's comments that our revised manuscript is more complete and has improved in quality. We also thank the reviewer's point that *Cosmc* deletion affects B cell development in the bone marrow, as shown in Fig.2, which might impact the observed homing defects and should be interpreted carefully. Immature B cells exit from bone marrow and migrate to spleen to further mature before circulating to other lymphoid organs. In our case, as shown in Fig.2C and 2D, *Cosmc*-deficient B cells seem still able to reach full maturation, as observed by substantial proportions of B cell subsets detected in the spleen of BC-*Cosmc*KO mice. To address the reviewer's comments, we did not rule out the possibility that the mature *Cosmc*-deficient B cells might not be able to efficiently recirculate back to the bone marrow. Thus, we have stated in the main text that "The *Cosmc*-deficient mature B cells were substantially reduced in the bone marrow, compared to those in the spleen, indicating a possibly impaired recirculation of mature B cells back to the bone marrow." With current data, we think the profound feature of *Cosmc*-deficient B cells is their markedly reduced ability to migrate to lymphoid and nonlymphoid organs. We appreciate author's suggestion to consider choosing a different title of this manuscript. Cumulatively, based on all the data that we have presented in this study, we feel comfortable with the current title for its succinctness and emphasis on the main discovery, and we hope that the reviewer will concur.

2. What is the time frame of the intravital movies? Which time span was recorded? Can the authors add a time stamp to the movies? Along that line, what is the size of the field of view in the intravital movies? The authors should add some kind of reference, such as a scale bar. A movie legend is missing and should be added.

Typos:

L 94 B cell not B cells

L 95 block instead of blocks

Author Response: We thank the reviewer for this comment. We recorded the cells for 20 minutes immediately following injection of the cells into the left femoral artery. The movies have been updated with the timestamp as suggested. Unfortunately, we could not add a scale bar as

the software (CoreView version 2.0.2.13) does not have a feature to add a scale bar (confirmed from the manufacturer of the imaging system (IO Industries)). The imaged field is 1,339 μm x 1,331 μm . This information is now added to the movie legend.

We also thank the reviewer for pointing out the two typos and we have corrected them in the main text as follows: “we generate the murine B cell-specific *Cosmc* KO mice, which specifically block extension of O-GalNAc type O-glycans on glycoproteins of B cells.”

Reviewer #2 (Remarks to the Author):

The authors have followed the suggestions and significantly improved the manuscript.

Author Response: We thank the reviewer for the comment that our manuscript has been significantly improved.

Reviewer #3 (Remarks to the Author):

After having carefully studied the revised manuscript I come to the following conclusions:

1) Contrary to common practice, the authors do not provide a manuscript in which the revised passages are marked in change-track mode. It is therefore rather difficult to identify those changes that have been made to the text and to figures.

Author Response: We would like to apologize for the inconvenience; the tracked change file was provided previously, but must not have reached you. In this revised version of the manuscript, we have used track changes mode as suggested.

*2) One of my key points when reviewing the initial manuscript was that the study was rather preliminary and that further investigations should be done. In particular I suggested investigating into more detail the mechanisms underlying impaired homing of B cells via HEVs. Unfortunately this was not done. In their rebuttal letter, the authors write that such studies are currently not possible and that new experimental procedures need to be established. This is very surprising, however, as John Kehrl's group used multiphoton microscopy in a manuscript recently published in *iScience* to study in detail homing of lymphocytes via HEV (doi: 10.1016/j.isci.2019.05.040). That study actually provides a blueprint of the experiments needed here to gain further insights at what stage of their journey through HEV *Cosmc*-deficient B cells get stuck and how their behavior differs from that of their wild type counterpart.*

Author Response: We thank the reviewer for critical comments and suggestions. We agree with this reviewer's comment that a more detailed study on the mechanism underlying impaired homing of B cells via HEV is important. We thank the reviewer for highlighting one of the important directions that we are interested in pursuing in such studies. Also, we agree with the reviewer that applying the technique that was developed by John Kehrl's group, and having potentially some meaningful behavioral nature of *Cosmc*-deficient B cells journey through HEV compared to WT would be informative. Along this line, we have acknowledged this publication and updated the main text as follows: “In addition, recent notable technical advances have been made in visualizing how normal lymphocytes undergo transendothelial migration⁵⁵. Such novel methodology, which is beyond the scope of our present discovery, may reveal more details about the behavioral changes of *Cosmc*-deficient B cells during their migration through HEVs.” However, from our perspective, as we discussed in the main text, defining the O-glycosylated proteins and molecules on B cells and their binding partners on the endothelium will provide

more critical information to further dissecting how loss of O-glycans alters the B cell homing pattern. In this regard, we have had to develop new methodology and will report that separately.

3) Was ref 14 (Cutler et al) already mentioned in the first version of the manuscript? I do not remember that the authors had adequately addressed impaired homing of cosmc^{-/-} T cells there. Anyway! In Cutler et, it was shown that T cell-specific deletion of cosmic leads to impaired homing of T cells. In the present manuscript analogous results are shown for B cells without appropriately elucidating the underlying mechanisms. The authors speculate that the function of chemokine receptors may be impaired. However, only transwell assays were performed but no further analysis is shown.

In summary, I maintain my opinion that the B cell-specific cosmc^{-/-} mice show an interesting homing phenotype, but the results presented here are too preliminary for possible publication in Nature Communications.

Author Response: We thank this reviewer's constructive criticisms. Ref 14 (Cutler et al) was not in the previous version and we included it in the second version of the manuscript as we elaborated the background for our current study. Lymphocyte subset-specific migration pattern has been observed over the past years. We suspect that O-glycans may exert their functions in T and B cells' migration through differential mechanisms. We certainly will explore further the mechanisms by which O-glycans regulate lymphocyte homing. However, as we pointed out in our first rebuttal, new methodologies are needed to be developed and optimized first. Overall, we believe that our current study has provided a novel perspective in understanding the key importance of O-glycosylation in the intricate multistep processes of lymphocyte migration. Our finding will accelerate studies in this relatively new direction, and as more details are uncovered, the roles of surface glycans in lymphocytes in their migration and distribution will be clearer over time.